# Dimension reduction via score ratio matching

**Ricardo Baptista**[*]                                                        *rsb@caltech.edu*
*California Institute of Technology*
*Pasadena, CA 91125 USA*

**Michael C. Brennan**[*]                                                      *mcbrenn@mit.edu*
*Massachusetts Institute of Technology*
*Cambridge, MA 02139 USA*

**Youssef Marzouk**                                                            *ymarz@mit.edu*
*Massachusetts Institute of Technology*
*Cambridge, MA 02139 USA*

**Reviewed on OpenReview:** *https://openreview.net/forum?id=mvbZBaqSXo*

## Abstract

Gradient-based dimension reduction decreases the cost of Bayesian inference and probabilistic modeling by identifying maximally informative (and informed) low-dimensional projections of the data and parameters, allowing high-dimensional problems to be reformulated as cheaper low-dimensional problems. A broad family of such techniques identify these projections and provide error bounds on the resulting posterior approximations, via eigendecompositions of certain diagnostic matrices. Yet these matrices require gradients or even Hessians of the log-likelihood, excluding the purely data-driven setting and many problems of simulation-based inference. We propose a framework, derived from score-matching, to extend gradient-based dimension reduction to problems where gradients are unavailable. Specifically, we formulate an objective function to directly learn the score ratio function needed to compute the diagnostic matrices, propose a tailored parameterization for the score ratio network, and introduce regularization methods that capitalize on the hypothesized low-dimensional structure. We also introduce a novel algorithm to iteratively identify the low-dimensional reduced basis vectors more accurately with limited data based on eigenvalue deflation methods. We show that our approach outperforms standard score-matching for problems with low-dimensional structure, and demonstrate its effectiveness for PDE-constrained Bayesian inverse problems and conditional generative modeling.

## 1 Introduction

A central aim of Bayesian computation is to develop efficient algorithms for characterizing conditional distributions, e.g., the posterior $\pi_{X|Y=y^*}$ of a parameter $X \in \mathbb{R}^n$, given an observation $y^* \in \mathbb{R}^m$ and their joint distribution $\pi_{X,Y}$. The cost of such procedures can become prohibitive with growing $n$ and $m$, making inference difficult for high (or possibly infinite)-dimensional problems. One approach for mitigating this computational burden is dimension reduction.

In this paper, we are interested in two types of low-dimensional structure that appear in many Bayesian inference problems, and more generally in probabilistic modeling. First is the notion that the target distribution can be well approximated as a *low-dimensional update* of a dominating reference distribution. Second is the notion that the conditioning variables can be replaced with *low-dimensional projections* or summaries. A recent line of work (Brennan et al., 2020; Zahm et al., 2022; Cui & Zahm, 2021; Baptista et al., 2022; Cui & Tong, 2022) has developed *gradient-based* methods for identifying and exploiting both

---

[*]Authors contributed equally to this work.

types of low-dimensional structure in non-Gaussian settings. These approaches construct specific *diagnostic matrices* containing averaged gradient or Hessian information of the posterior or joint log-density. Eigen-decompositions of these diagnostic matrices reveal informative projections of the conditioning variables and informed projections of the parameters, and also yield bounds on the error of the resulting approximations to the posterior in terms of the rank of these projections. Specifically, the spectra of these matrices indicate to what extent a given problem has the posited low-dimensional structure.

Here we propose a framework derived from *score-matching* to extend these ideas to *problems where gradients are unavailable.* In particular, we introduce a learning problem to approximate the gradient of the log-ratio of two densities, which we term a *score ratio* function. We show that score ratios are central objects within the diagnostic matrices described above, and that learning the score ratio can itself take advantage of low-dimensional structure in the problems at hand. Doing so yields novel architectures and algorithms that outperform standard score-matching approaches, and ultimately enables more efficient approximations of the targeted conditional distributions. Our main contributions are as follows:

1. We propose algorithms for uncovering two types of low-dimensional structure in probabilistic models (low-dimensional updates of a reference measure and low-dimensional conditioning) based on score ratio matching (§3).

2. We introduce a novel training objective, network parameterizations, and regularization methods tailored to our dimension reduction goals (§4).

3. We develop a new algorithm that iteratively identifies a basis for the desired reduced subspaces more accurately with limited data (§5).

4. We demonstrate that our score ratio matching method better reveals low-dimensional structure compared to standard score matching, and that it enables more accurate and efficient high-dimensional approximate inference (§6).

**Related work:** Many previous papers have highlighted the benefits of gradient-based dimension reduction in Bayesian computation, for methods ranging from MCMC (Cui & Tong, 2022; Cui et al., 2014; Constantine et al., 2016) to SVGD (Chen & Ghattas, 2020) to normalizing flows (Brennan et al., 2020; Cui et al., 2023; Radev et al., 2020) to ensemble filtering (Le Provost et al., 2022). The present work is concerned with realizing these benefits in gradient-free settings, which include Bayesian inverse problems with complex forward models (Kaipio & Somersalo, 2006; Biegler et al., 2010), goal-oriented inference (Lieberman & Willcox, 2013; Berger et al., 1999), simulation-based inference (SBI) more broadly (Cranmer et al., 2020), and the purely data-driven setting where only a fixed set of samples from $\pi_{X,Y}$ are given.

The second form of dimension reduction we seek (low-dimensional conditioning) is related to SBI-focused efforts of creating summary (ideally sufficient) statistics of the observations to improve the quality and efficiency of inference algorithms (Fearnhead & Prangle, 2012; Joyce & Marjoram, 2008; Nunes & Balding, 2010). So far, most of these methods have not been gradient-based, and do not explicitly take advantage of low-dimensional structure when learning statistics. For instance, Radev et al. (2020) learns an (arbitrary) summary neural network to compress the observation variable before it enters an inference network. Brehmer et al. (2020); Alsing et al. (2018) define data summaries based on locally sufficient statistics, but only around one reference parameter value. More generally, hand-crafted summary statistics are often the norm in SBI (Sisson et al., 2018). These statistics often require expert knowledge to form or use specific selection criteria that are not tied to posterior quality, and hence may lead to issues with "poor quality features," as seen in Lueckmann et al. (2017). In contrast, the approach proposed here seeks optimal summaries that provide error guarantees on the resulting posterior approximation.

## 2 Background

### 2.1 Gradient-based dimension reduction

We provide a brief description of two gradient-based dimension reduction methods for which we will develop gradient-free versions. These methods are *data-free certified dimension reduction* (CDR)[1] (Cui & Zahm, 2021) and *conditional-mutual information based dimension reduction* (CMIDR) (Baptista et al., 2022). Both methods decompose the parameter and observation into two subspaces,

$$X = U_r X_r + U_\perp X_\perp \quad \text{where} \quad X_r = U_r^\top X, \ X_\perp = U_\perp^\top X \tag{1}$$

$$Y = V_s Y_s + V_\perp Y_\perp \quad \text{where} \quad Y_s = V_s^\top Y, \ Y_\perp = V_\perp^\top Y \tag{2}$$

where $U = [U_r \ U_\perp] \in \mathbb{R}^{n \times n}$ and $V = [V_s \ V_\perp] \in \mathbb{R}^{m \times m}$ are unitary matrices, and $U_r \in \mathbb{R}^{n \times r}$, $r \leq n$, and $V_s \in \mathbb{R}^{m \times s}$, $s \leq m$.

Intuitively each method seeks transformations $U$ and $V$ such that the projected variables $X_r$ and $Y_s$ capture where the parameter is most informed and where the observation is most informative, respectively. Given a decomposition in (1)–(2), Definition 2.1 below introduces a low-dimensional approximation of the posterior that departs from a known reference distribution using low-dimensional subspaces for the parameter and for the observation.

**Definition 2.1.** *Let $\rho$ be a chosen reference density on $\mathbb{R}^n$. Given unitary matrices $U \in \mathbb{R}^{n \times n}$ and $V \in \mathbb{R}^{m \times m}$, and integers $r \leq n$, $s \leq m$, let $\mathcal{D}_{r,s}(U, V)$ denote a set of distributions with densities of the form*

$$\widetilde{\pi}_{X|Y}(x|y) \propto f(U_r^\top x, V_s^\top y)\rho(x),$$

*for some $f \colon \mathbb{R}^{r+s} \to \mathbb{R}_{>0}$, where $U_r \in \mathbb{R}^{n \times r}$ contains the first $r$ columns of $U$ and $V_s \in \mathbb{R}^{m \times s}$ contains the first $s$ columns of $V$. The class of distributions where only parameter dimension reduction is considered is denoted $\mathcal{D}_{r,m}(U)$, and the class where only observation reduction is considered is denoted $\mathcal{D}_{n,s}(V)$.*

Propositions 2.2 and 2.3 below are key results for selecting the optimal transformations $U$ and $V$, respectively. The transformations arise from minimizing an error bound for the resulting posterior approximation of the form in Definition 2.1. To derive the error bound, Proposition 2.3 assumes that the joint distribution $\pi_{X,Y}$ satisfies a *subspace log-Sobolev inequality*; we provide a definition in Appendix A.1. We note that multivariate Gaussian distributions, Gaussian mixtures, and uniform distributions on compact and convex domains all satisfy a subspace log-Sobolev inequality; see Zahm et al. (2022, Assumption 2.5) for more discussion. Conditions on the forward model of an inverse problem that are sufficient for $\pi_{X,Y}$ to satisfy a subspace log-Sobolev inequality can be found in Baptista et al. (2022, Example 2).

**Proposition 2.2** (Modified from Section 3.3 of Cui & Zahm (2021))**.** *Let $\rho$ be the standard Gaussian density, and define the parameter diagnostic matrix*

$$H_{\mathrm{CDR}}^X = \mathbb{E}_{\pi_{X,Y}} \left[ \nabla_x \log \left( \frac{\pi_{X|Y}(x|y)}{\rho(x)} \right) \nabla_x \log \left( \frac{\pi_{X|Y}(x|y)}{\rho(x)} \right)^\top \right] \in \mathbb{R}^{n \times n}. \tag{3}$$

*Let $(\lambda_k^X, u_k) \in \mathbb{R}_{\geq 0} \times \mathbb{R}^n$ be the $k$-th eigenpair of $H_{\mathrm{CDR}}^X$, with $\lambda_1^X \geq \cdots \geq \lambda_n^X$, and take $U = [u_1, \ldots, u_n]$. Then there exists $\widetilde{\pi}_{X|Y} \in \mathcal{D}_{r,m}(U)$ such that*

$$\mathbb{E}_{\pi_Y} \left[ D_{KL}(\pi_{X|Y} || \widetilde{\pi}_{X|Y}) \right] \leq \frac{1}{2} \mathrm{Tr} \left( (I - U_r U_r^\top) H_{\mathrm{CDR}}^X \right) = \frac{1}{2} \sum_{k > r} \lambda_k^X. \tag{4}$$

**Proposition 2.3** (Modified from Theorem 1 of Baptista et al. (2022)[2])**.** *Define the observation diagnostic matrix*

$$H_{\mathrm{CMI}}^Y = \mathbb{E}_{\pi_{X,Y}} \left[ \nabla_x \nabla_y \log \left( \frac{\pi_{X,Y}(x,y)}{\rho(x)} \right)^\top \nabla_x \nabla_y \log \left( \frac{\pi_{X,Y}(x,y)}{\rho(x)} \right) \right] \in \mathbb{R}^{m \times m} \tag{5}$$

---

[1]We drop 'data-free' in the acronym as we will not consider the dimension reduction problem initially presented in Zahm et al. (2022) for the posterior corresponding to a single realization of the observation.

[2]While Baptista et al. (2022) present joint parameter and observation reduction results, we focus here on observation dimension reduction for ease of presentation.

*Assume that $\pi_{X,Y}$ satisfies the subspace logarithmic Sobolev inequality with constant $C(\pi_{X,Y})$. Let $(\lambda_k^Y, v_k) \in \mathbb{R}_{\geq 0} \times \mathbb{R}^m$ be the $k$-th eigenpair of $H_{\mathrm{CMI}}^Y$ with $\lambda_1^Y \geq \cdots \geq \lambda_m^Y$ and take $V = [v_1, \ldots, v_m]$. Then, there exists $\widetilde{\pi}_{X|Y} \in \mathcal{D}_{n,s}(V)$ such that*

$$\mathbb{E}_{\pi_Y}\left[D_{KL}(\pi_{X|Y}||\widetilde{\pi}_{X|Y})\right] \leq C(\pi_{X,Y})^2 \operatorname{Tr}\left((I - V_s V_s^\top)H_{\mathrm{CMI}}^Y\right) = C(\pi_{X,Y})^2 \sum_{k>s} \lambda_k^Y. \tag{6}$$

Note that there is no explicit assumption of a subspace log-Sobolev inequality in Proposition 2.2 because, in the setting of parameter dimension reduction, we need this assumption only on the reference measure $\rho$, and since $\rho$ here is chosen to be standard Gaussian, the inequality is satisfied with a log-Sobolev constant of one.

Propositions 2.2 and 2.3 have several practical implications. Given the diagnostic matrices $H_{\mathrm{CDR}}^X$ and $H_{\mathrm{CMI}}^Y$, we should choose $U_r$ and $V_s$ as the leading eigendirections of these matrices. (As shown in Zahm et al. (2022), solving these eigenvalue problems minimizes a more general upper bound.) With this choice, we also have upper bounds on the approximation error (in KL divergence) incurred by parameter and observation reduction, which can be used to select the reduced dimensions $r$ and $s$.

## 2.2 Approximating score functions

Score matching has recently appeared as a powerful unsupervised learning framework with applications to generative modeling (Song & Ermon, 2019; Song et al., 2021; Ho et al., 2020; De Bortoli et al., 2021) and Bayesian inference (Zhang et al., 2018; Pacchiardi & Dutta, 2022). The core task is to approximate the *score function*, the gradient of a log-density function, for various downstream tasks. We direct readers to Song & Ermon (2019); Song et al. (2021) for an overview of score matching, especially the derivation of the objective function for learning the score, network training strategies, and its application to generative modeling using Langevin sampling. In this section, we focus on conditional score-matching, where one approximates the score function of the conditional distribution $\nabla_x \log \pi_{X|Y}(x|y)$.

Let $w_\theta : \mathbb{R}^{n+m} \to \mathbb{R}^n$ be the neural network approximation of $\nabla_x \log \pi_{X|Y}(x|y)$, were $\theta$ denotes the learnable parameters of the neural network. Score matching seeks to minimize a weighted $L^2$ error between the true and approximate scores

$$J^*(w_\theta) = \frac{1}{2}\mathbb{E}_{\pi_{X,Y}}\|\nabla_x \log \pi_{X|Y}(x|y) - w_\theta(x,y)\|^2.$$

While minimizing this objective is intractable as it requires access to the score function, approaches known as implicit score matching (Hyvärinen & Dayan, 2005) and denoising score matching (Vincent, 2011) introduce reformulated objective functions for $J^*(\theta)$ to learn the score based only on *samples* from a given distribution, such as $\pi_{X,Y}$. We will focus on implicit score matching in this work, although this is not a limitation. Under mild regularity assumptions on the true and approximate score functions (see Assumption 3.1), Hyvärinen & Dayan (2005) showed that the objective $J^*$ can be written as

$$J^*(w_\theta) = \mathbb{E}_{\pi_{X,Y}}\left[\frac{1}{2}\|w_\theta(x,y)\|^2 + \operatorname{Tr}(\nabla_x w_\theta(x,y))\right] + C,$$

where $C$ is a constant with respect to the score network. This objective only depends on $\pi$ via the expectation, and so in practice one can learn $w_\theta(x,y)$ by minimizing an objective that estimates the expectation using joint parameter-observation samples $\{x^{(j)}, y^{(j)}\}_{j=1}^N \sim \pi_{X,Y}$. That is,

$$J(w_\theta) = \sum_{j=1}^N \frac{1}{2}\|w_\theta(x^{(j)}, y^{(j)})\|^2 + \operatorname{Tr}(\nabla_x w_\theta(x^{(j)}, y^{(j)})).$$

As discussed in Song et al. (2020), directly evaluating the term $\operatorname{Tr}(\nabla_x w_\theta(x^{(j)}, y^{(j)}))$ in the objective function is prohibitively expensive for even moderate dimensions $n$. In practice, this is alleviated using the Hutchinson's trace estimator (Hutchinson, 1989).

# 3 Score ratio matching

Here we propose a tailored score matching approach to construct the diagnostic matrices in Section 2.1. To do so, we consider approximations of the *score ratio* function

$$w(x, y) \coloneqq \nabla_x \log \left( \pi_{X|Y}(x|y)/\rho(x) \right),$$

where $\rho$ is chosen to be a tractable reference density (e.g., a standard normal) and present several results related to the dimension reduction approaches of Section 2.1. First we note that both diagnostic matrices $H_{\text{CDR}}^X$ and $H_{\text{CMI}}^Y$ can be expressed in terms of the score ratio function. Indeed, $H_{\text{CDR}}^X$ is defined explicitly in terms of $w(x, y)$, while $H_{\text{CMI}}^Y$ depends on the mixed gradient function

$$\nabla_x \nabla_y \log \left( \pi_{X,Y}(x, y)/\rho(x) \right) = \nabla_y w(x, y).$$

Therefore, approximating the score ratio allows us to construct the two diagnostic matrices and perform dimension reduction of both parameters and observations.

A naïve strategy would be to approximate $\nabla_x \log \pi_{X|Y}(x|y)$ and use it to compute the score ratio as the difference,

$$\nabla_x \log(\pi_{X|Y}(x|y)/\rho(x)) = \nabla_x \log \pi_{X|Y}(x|y) - \nabla_x \log \rho(x).$$

Instead, we take a different approach that leverages the (possible) low-dimensional structure of $\pi_{X|Y}$. Under the hypothesis that $\pi_{X|Y}$ is well approximated within $\mathcal{D}_{r,s}(U, V)$ for some choices of $U$ and $V$, we expect the score ratio, rather than the score itself, to be well approximated by a *ridge function* (Pinkus, 2015), i.e., a function that is constant for $x, y \in \text{Im}(U_\perp) \times \text{Im}(V_\perp)$. In Section 4, we describe a parameterization of $w_\theta(x, y)$ and a regularization method that are tailored to learning this low-dimensional structure.

Theorem 3.2 provides an objective function that allows for direct approximation of the score ratio using a *score ratio network* $w_\theta \colon \mathbb{R}^{n+m} \to \mathbb{R}^n$. This and subsequent results rely on the following mild assumptions on the posterior and the score ratio approximation.

**Assumption 3.1.** *Let $w_\theta$ denote the score (or score ratio) approximation. We assume (i) $w_\theta$ is differentiable with respect to $x$; (ii) $\pi_{X|Y}(x|y)$ is differentiable with respect to $x$; (iii) $\mathbb{E}_{\pi_{X,Y}} \|w_\theta\|^2 < \infty$ for all $\theta$; (iv) $\mathbb{E}_{\pi_{X,Y}} \|\nabla_x \log(\pi_{X|Y}(x|y)/\rho(x))\|^2 < \infty$; and (v) $\pi_{X|Y}(x|y)w_\theta \to 0$ as $\|x\| \to \infty$ for all $\theta$ and $y$.*

**Theorem 3.2.** *Let $w_\theta \colon \mathbb{R}^{n+m} \to \mathbb{R}^n$ be the score ratio approximation. Under the conditions of Assumption 3.1, we have the following equivalence of objectives:*

$$\frac{1}{2} \mathbb{E}_{\pi_{X,Y}} \left\| w_\theta(x, y) - \nabla_x \log \left( \frac{\pi_{X|Y}(x|y)}{\rho(x)} \right) \right\|_2^2$$

$$= \mathbb{E}_{\pi_{X,Y}} \left[ \frac{1}{2} w_\theta(x, y)^\top w_\theta(x, y) + \text{Tr}(\nabla_x w_\theta(x, y)) + \nabla_x \log \rho(x)^\top w_\theta(x, y) \right] + C, \tag{7}$$

*where $C$ is a constant that only depends on the densities $\pi_{X|Y}$ and $\rho$.*

We give a proof of the theorem in Appendix A.2. As in implicit score-matching, Theorem 3.2 shows that the approximation can be learned with an objective that does not explicitly depend on the true score ratio.

In practice, we replace the expectation on the right-hand side of (7) with a Monte Carlo estimate using joint samples $\{x^{(j)}, y^{(j)}\}_{j=1}^N$, and thus define the optimization objective,

$$J(w_\theta) \coloneqq \frac{1}{N} \left[ \sum_{j=1}^N \frac{1}{2} w_\theta(x^{(j)}, y^{(j)})^\top w_\theta(x^{(j)}, y^{(j)}) + \text{Tr}(\nabla_x w_\theta(x^{(j)}, y^{(j)})) + \nabla_x \log \rho(x^{(j)})^\top w_\theta(x^{(j)}, y^{(j)}) \right]. \tag{8}$$

Once we identify the score ratio approximation $w_\theta$ that minimizes this objective, the diagnostic matrices can be approximated as

$$H_{\text{CDR}}^X \approx \widehat{H}_{\text{CDR}}^X \coloneqq \mathbb{E}_{\pi_{X,Y}} \left[ w_\theta(x, y) w_\theta(x, y)^\top \right]$$

$$H_{\text{CMI}}^Y \approx \widehat{H}_{\text{CMI}}^Y \coloneqq \mathbb{E}_{\pi_{X,Y}} \left[ \nabla_y w_\theta(x, y)^\top \nabla_y w_\theta(x, y) \right].$$

Estimators for the parameter and observation transformations $U$ and $V$ are then given by the leading eigenvectors of $\widehat{H}^X_{\mathrm{CDR}}$ and $\widehat{H}^Y_{\mathrm{CMI}}$, respectively. Given a reduced posterior that is constructed using the estimated transformations, a question that naturally arises is how the accuracy of the reduced posterior is affected by error in our approximation of the score ratio. The follow theorem provides an answer specifically for CDR.

**Theorem 3.3.** *Let $w_\theta(x, y)$ be an approximation to the score ratio satisfying*

$$\mathbb{E}_{\pi_{X,Y}} \left\| w_\theta(x, y) - \nabla_x \log \left( \frac{\pi_{X|Y}(x|y)}{\rho(x)} \right) \right\|^2 \le \epsilon.$$

*Define the approximate parameter diagnostic matrix $\widehat{H}^X_{\mathrm{CDR}} = \mathbb{E}_{\pi_{X,Y}} \left[ w_\theta(x,y) w_\theta(x,y)^\top \right]$. Let $(\lambda_i, u_i) \in \mathbb{R}_{\ge 0} \times \mathbb{R}^n$ be the $i$-th eigenpair of $\widehat{H}^X_{\mathrm{CDR}}$, with $\lambda_1 \ge \cdots \ge \lambda_n$, and take $U = [u_1, \ldots, u_r]$. Then there exists a $\widetilde{\pi}_{X|Y} \in \mathcal{D}_{r,m}(U)$ such that*

$$\mathbb{E}_{\pi_Y} \left[ D_{KL}(\pi_{X|Y} || \widetilde{\pi}_{X|Y}) \right] \le \epsilon + \sum_{k > r} \lambda_k.$$

See Appendix A.3 for the proof. We note that the approximation error of the score ratio network $\epsilon$ is generally unknown in practice. Theorem 3.3, however, establishes stability of the error in the approximate posterior distribution $\widetilde{\pi}_{X|Y}$, which depends on the subspace $U$, with respect to error in the score ratio used to identify that subspace.

**Choice of the KL divergence as the error metric**  We note that our methods specifically seek projections of the parameter and observation that minimize bounds on the posterior approximation error measured by the data-averaged KL divergence $\mathbb{E}_{\pi_Y} \left[ D_{KL}(\pi_{X|Y} || \widetilde{\pi}_{X|Y}) \right]$, directly following the results of Propositions 2.2 and 2.3. While the data-averaged KL divergence is a natural and widely used choice, alternative error metrics such as a worst-case KL divergence (over $Y$), Wasserstein distances, or total variation distances may provide additional insights or be better suited for specific inference tasks. Developing methods that minimize these alternative metrics and studying the differences in the posterior approximations could be an interesting direction for future work, particularly in cases where robustness to tail behavior or accuracy of specific posterior expectations are important.

**Comparison with other common dimension reduction methods**  Here we comment on the differences between the present work and several common dimension reduction methods. The classical method of principal component analysis (PCA) seeks directions that maximize the variance of the data. Nonlinear methods such as t-SNE (Van der Maaten & Hinton, 2008) and UMAP (McInnes et al., 2018) can effectively preserve local neighborhood structures, making them powerful tools for visualization and marginal dimension reduction. However, these methods do not explicitly consider the *joint* structure between parameters and observations—which is crucial for inference problems, where preserving conditional relationships is essential. We reiterate that our framework seeks projections of the parameter and observation that are informed and informative, respectively, making it particularly suited to inference. In comparison, methods that act *marginally* (e.g., performing PCA, t-SNE, UMAP etc. on $X$ and $Y$ individually) do not capture the dependence structure between the parameter and observation and thus do not explicitly consider the underlying inference problem.

We note that in the case of linear-Gaussian likelihood models, the projections proposed by Propositions 2.2 and 2.3 are equivalent to those found by canonical correlation analysis (CCA) based on covariance information between $X$ and $Y$; see Proposition 5 in Baptista et al. (2022). Insofar as the gradient-based methods proposed in Baptista et al. (2022) can be viewed as a generalization of CCA to non-Gaussian likelihood models (e.g., by using gradient information of a possibly nonlinear forward model), our methods further generalize to problems where gradients of these likelihood functions are not available.

**Choice of reference distribution**  We emphasize that the choice of the reference distribution in our method is a degree of freedom, provided that its score function is easily evaluated. Our method identifies directions in parameter space where the posterior differs from the reference distribution on average, and is

most effective when these directions are contained within a low-dimensional subspace. The marginal distribution of the parameter (i.e., the prior in Bayesian settings) is thus the natural first choice, when tractable. In cases where the prior can be easily mapped to a standard Gaussian, we recommend reparameterizing the problem accordingly for computational convenience (as done in Sections 6.1 and 6.2). However, when such a "whitening" is not easily performed, as in Sections 6.3 and 6.4, we suggest using a standard Gaussian reference distribution after centering and scaling the parameter. Future work will explore other methods for picking the reference distribution when the parameter marginal is not available.

## 4  Structure-exploiting networks and regularization

We now describe a parameterization for $w_\theta(x, y)$ and a regularization method that uncover possible low-dimensional structure in the target distribution. Recall that for $\widetilde{\pi}_{X|Y} \in \mathcal{D}_{r,s}(U, V)$, we have

$$\widetilde{\pi}_{X|Y}(x|y) \propto f(U_r^\top x, V_s^\top y)\rho(x),$$

for some function $f \colon \mathbb{R}^{r+s} \to \mathbb{R}_{>0}$, where $U_r \in \mathbb{R}^{n \times r}$ contains the first $r$ columns of $U$, and $V_s \in \mathbb{R}^{m \times r}$ contains the first $s$ columns of $V$. Then we note that the score ratio and its observation gradient take the specific form

$$\nabla_x \log\left(\frac{\widetilde{\pi}_{X|Y}(x|y)}{\rho(x)}\right) = U_r g(U_r^\top x, V_s^\top y), \quad \nabla_y \nabla_x \log\left(\frac{\widetilde{\pi}_{X|Y}(x|y)}{\rho(x)}\right) = U_r h(U_r^\top x, V_s^\top y)V_s^\top,$$

where $g(x_r, y_s) \coloneqq \nabla_{x_r} \log f(x_r, y_s)$, and $h(x_r, y_s) \coloneqq \nabla_{y_s} \nabla_{x_r} \log f(x_r, y_s)$. Observe that the range of the score ratio lies within the subspace spanned by $U_r$, and the range of $\nabla_x \nabla_y \log(\widetilde{\pi}_{X|Y}(x|y)/\rho(x))^\top$ lies within the subspace spanned by $V_s$. We encode these structural ansatzes into the parameterization of the score ratio network

$$w_\Theta(x, y) = W_x \psi_\theta(W_x^\top x, W_y^\top y),$$

where $W_x \in \mathbb{R}^{n \times r'}$, $W_y \in \mathbb{R}^{m \times s'}$, $\psi_\theta \colon \mathbb{R}^{r'+s'} \to \mathbb{R}^{r'}$ is a typical conditional score network for some $r' \le n$ and $s' \le m$, and $\Theta = (\theta, W_x, W_y)$ denotes all trainable model parameters. If $W_x$ and $W_y$ converge toward matrices that have low (effective) ranks (much smaller than $n$ and $m$) during optimization, the ranges of $w_\Theta(x, y)$ and $\nabla_y w_\Theta(x, y)$ are restricted accordingly. We note that this parameterization also allows us to constrain the highest possible ranks of the estimated diagnostic matrices. That is, the rank of the parameter diagnostic matrices is bounded by $r'$ and the rank of the observation diagnostic matrix is bounded by $s'$.

To promote $W_x$ and $W_y$ be low-rank when $\pi_{X|Y}$ is expected to have low-dimensional structure, we penalize the *nuclear norms* of $W_x$ and $W_y$ during optimization, as is in commonly used for low-rank matrix estimation (Fazel, 2002). This leads to our final objective function

$$\mathcal{J}(\Theta) \coloneqq J(w_\Theta) + \lambda_1 \|W_x\|_* + \lambda_2 \|W_y\|_*$$

where $\|\cdot\|_*$ is the nuclear norm, $\lambda_1, \lambda_2 \ge 0$ are regularization parameters, and $J(w_\Theta)$ is the objective function defined in (8). Algorithm 1 presents the complete score ratio matching procedure to estimate the parameter and observation transformations as well as the dimensions of the reduced variables based on a specified tolerance for the posterior approximation error.

## 5  Deflating score-ratio matching

Given that dimension reduction acts as a pre-processing step before performing inference, we wish to reduce the cost of constructing the diagnostic matrices as much as possible. To this end, we would like to use relatively small sets of training samples and small networks that can be optimized easily. For problems with large parameter and observation dimensions, we find that a given score ratio network often estimates diagnostic matrices that accurately capture the leading eigenvectors of the true diagnostic matrix, but that higher-indexed eigenvectors are inaccurate.

---

**Algorithm 1** Single network score ratio dimension reduction

---

1: **Input**: Target data $\{x^{(j)}, y^{(j)}\}_{j=1}^N \sim \pi_{X,Y}$, and user tolerances $\varepsilon_X, \varepsilon_Y > 0$
2: Solve

$$\min_{\theta, W_x, W_y} \mathcal{J}(\theta, W_x, W_y)$$

to obtain the score-ratio approximation $w_\theta(x, y)$.
3: Estimate the diagnostic matrices

$$\widehat{H}_{\mathrm{CDR}}^X = \frac{1}{N} \sum_{j=1}^N w_\theta(x^{(j)}, y^{(j)}) w_\theta(x^{(j)}, y^{(j)})^\top$$

$$\widehat{H}_{\mathrm{CMI}}^Y = \frac{1}{N} \sum_{j=1}^N \nabla_y w_\theta(x^{(j)}, y^{(j)})^\top \nabla_y w_\theta(x^{(j)}, y^{(j)})$$

4: Compute the eigenpairs: $(\lambda_i^X, \widetilde{u}_i) \in \mathbb{R}_{\geq 0} \times \mathbb{R}^n$ of $\widehat{H}_{\mathrm{CDR}}^X$ and $(\lambda_i^Y, \widetilde{v}_i) \in \mathbb{R}_{\geq 0} \times \mathbb{R}^m$ of $\widehat{H}_{\mathrm{CMI}}^Y$.
5: Pick $r$ and $s$ so that

$$\frac{1}{2} \sum_{k>r} \lambda_k^X < \varepsilon_X, \quad \sum_{k>s} \lambda_k^Y < \varepsilon_Y$$

and set $\widetilde{U}_r = [\widetilde{u}_1 \ \ldots \ \widetilde{u}_r]$, $\widetilde{V}_s = [\widetilde{v}_1 \ \ldots \ \widetilde{v}_s]$.
6: **output:** $\widetilde{U}_r, \widetilde{V}_s$

---

To improve the estimation of more eigenvectors of the two diagnostic matrices, we propose an iterative method inspired by eigenvalue deflation methods (see Saad (2011, Chapter 4) for a comprehensive review). Rather than finding all of the columns of $U_r$ and $V_s$ using a single score ratio network, we construct a sequence of score ratio networks designed to capture progressively more vectors that form the transformations.

The following lemma defines a deflated matrix that can be used to reveal higher-index eigenvectors. Then, Proposition 5.2 shows how to construct a modified score ratio to compute deflated diagnostic matrices.

**Lemma 5.1.** *Let $(\lambda_i, \phi_i)$, for $i = 1, \ldots, d$, be the eigenpairs of a symmetric matrix $A \in \mathbb{R}^{d \times d}$, with $\lambda_1 > \lambda_2 \geq \ldots \lambda_d$ and $\|\phi_i\|_2 = 1$. Define the subunitary matrix $\Phi_r = [\phi_1 \ \ldots \ \phi_r]$ and projector $P = I - \Phi_r \Phi_r^\top$. Then the deflated matrix $\widetilde{A} := PAP$ has eigenpairs $(0, \phi_i)$ for $i = 1, \ldots, r$ and $(\lambda_i, \phi_i)$ for $i = r+1, \ldots, d$.*

See Appendix A.4 for the proof.

**Proposition 5.2.** *Let $P^X \in \mathbb{R}^{n \times n}$ and $P^Y \in \mathbb{R}^{m \times m}$ be orthogonal projectors. Define the deflated score ratio*

$$w_P(x, y) = P^X \nabla_x \log \left( \frac{\pi_{X|Y}(x | P^Y y)}{\rho(x)} \right). \tag{9}$$

*Then the diagnostic matrices computed using $w_P$ take the form*

$$\widetilde{H}_{\mathrm{CDR}} := \mathbb{E}_{\pi_{X,Y}} \left[ w_P(x, y) w_P(x, y)^\top \right] = P^X H P^X$$

$$\widetilde{H}_{\mathrm{CMI}}^Y := \mathbb{E}_{\pi_{X,Y}} \left[ \nabla_y w_P(x, y)^\top \nabla_y w_P(x, y) \right] = P^X H' P^X,$$

*where $H \in \mathbb{R}^{n \times n}$ and $H' \in \mathbb{R}^{m \times m}$ are symmetric matrices. Notably, this implies $\ker(P^X) \subset \ker(\widetilde{H}_{\mathrm{CDR}})$ and $\ker(P^Y) \subset \ker(\widetilde{H}_{\mathrm{CMI}}^Y)$.*

See Appendix A.5 for the proof. Let $P^X$ and $P^Y$ be orthogonal projectors onto the span of a few previously computed eigenvectors, e.g., those computed from the diagnostic matrices estimated in Algorithm 1. Then, Proposition 5.2 allows us to learn a new score network that produces deflated diagnostic matrices whose leading eigenvectors are orthogonal to previously computed eigenvectors. This process can be then repeated by defining new projectors onto the span of a larger collection of eigenvectors. Algorithm 2 describes the complete numerical procedure using multiple deflation steps.

**Discussion on computational costs** Here, we discuss the computational costs and scalability of the methods described in Algorithms 1 and 2. First, we emphasize that our framework functions as a pre-inference step: we only require joint *prior* samples, $\{x^{(j)}, y^{(j)}\}_{j=1}^N \sim \pi_{X,Y}$. Also, rather than needing sufficient sample size and network expressivity to precisely recover the score (as one might want for posterior sampling), empirically we find that much less is needed to capture the low-dimensional structure underlying the problem, i.e., the dominant eigenspaces of the diagnostic matrices. This allows us to leverage that structure during inference, reducing the overall computational burden of the problem.

In general, approximating the score ratio encounters challenges similar to those of traditional score matching techniques. Assuming the score of the reference distribution can be evaluated efficiently, the computational cost of evaluating the traditional score matching loss and the loss function derived in Theorem 3.2 are the same up to a constant. That said, our network parameterization and regularization scheme are specifically designed to exploit low-dimensional structure in the target problem. As demonstrated in Section 6.1, our approach exhibits better sample efficiency than traditional score matching. The iterative deflating method (Algorithm 2) can further improve sample efficiency, as each score ratio function is restricted to learning a very low dimensional subspace.

---

**Algorithm 2** Iterative-deflated score ratio dimension reduction

1: **Input**: Target data $\{x^{(j)}, y^{(j)}\}_{j=1}^N \sim \pi_{X,Y}$, number of deflation steps $T$, number of eigenvectors to keep at each step $\ell \leq r', s'$.
2: Initialize deflating orthogonal projectors

$$P^X = I_n, \ P^Y = I_m$$

3: **for** $t = 1, \ldots, T$ **do**
4:      Parameterize the projected score ratio network in the form of eq. (9)
5:      Obtain the leading $\ell$ reduction vectors $\widetilde{U}_\ell$ and $\widetilde{V}_\ell$ via Algorithm 1
6:      Update the reduction basis vectors $\widetilde{U} \leftarrow [\widetilde{U} \ \widetilde{U}_\ell], \widetilde{V} \leftarrow [\widetilde{V} \ \widetilde{V}_\ell]$
7:      Update the orthogonal projectors

$$P^X \leftarrow P^X - \widetilde{U}_\ell \widetilde{U}_\ell^\top, \ P^Y \leftarrow P^Y - \widetilde{V}_\ell \widetilde{V}_\ell^\top$$

8: **end for**
9: **output:** $\widetilde{U}, \widetilde{V}$

---

## 6 Numerical examples

We now present several numerical experiments to show the utility of our methods. Table 1 in the Supplementary Material summarizes all network and training hyperparameter choices for each numerical example. For the problems presented in Sections 6.1 and 6.2, the score ratio is tractable and thus we can construct the true diagnostic matrices $H_{\mathrm{CDR}}^X$ and $H_{\mathrm{CMI}}^Y$. We thus evaluate the accuracy of our method by comparing the posterior approximation errors for the optimal basis transformations in (4) and (6) with the following error bounds achieved using the learned bases $\widetilde{U}$ and $\widetilde{V}$[3]:

$$E_r^{\mathrm{CDR}}(\widetilde{U}) := \frac{1}{2} \operatorname{Tr}((I - \widetilde{U}_r \widetilde{U}_r^\top) H_{\mathrm{CDR}}^X), \qquad E_s^{\mathrm{CMI}}(\widetilde{V}) := \operatorname{Tr}((I - \widetilde{V}_s \widetilde{V}_s^\top) H_{\mathrm{CMI}}^Y).$$

We emphasize that this analysis is meant to validate our method and is only possible when the true diagnostic matrices are computable. For the problems of Sections 6.3 and 6.4, the true diagnostic matrix is not computable. In these cases we validate our method by showing we achieve better inference fidelity for the reduced problems as compared to the non-reduced problems.

---

[3]We use a tilde to distinguish learned bases, i.e., eigenvectors of the diagnostic matrices computed using a score (ratio) network, from the eigenvectors of the exact diagnostic matrices.

### 6.1 Distribution with planted low-dimensional structure

First, we consider the "embedded banana" distribution where the data-generating process is given by

$$X_1' \sim \mathcal{N}(0,1), \quad X_2'|x_1' \sim \mathcal{N}(x_1'^2, 1), \quad X_{3:10}' \sim \mathcal{N}(0, I), \tag{10}$$

and $X = RX'$, where $R \in \mathbb{R}^{10 \times 10}$ is a random rotation matrix that is sampled by computing the QR factorization of a random matrix with standard Gaussian entries. In this case, the distribution $\pi_X$ is not a function of any observation, and so we only consider parameter dimension reduction. Given that $\pi_X$ only departs from a standard Gaussian along the coordinates $(x_1', x_2')$, we expect our algorithm to find the subspace spanned by the two leading columns of $R$. In this example, we are able to compute the score ratio analytically and directly estimate the true diagnostic matrix $H_{\mathrm{CDR}}^X$.

In Figure 1a we plot the error bound $E_r^{\mathrm{CDR}}$ for three different bases: (1) the eigenbasis of the true diagnostic matrix; (2) the eigenbasis of the diagnostic matrix computed with our score ratio approximation; and (3) the eigenbasis of the diagnostic matrix computed with a standard score approximation (i.e., a score network approximating $\nabla_x \log \pi_X(x)$). Both the score ratio and standard score networks were trained with $N = 1000$ samples. For our method, we see that the error bound sharply drops at $r = 2$ to less than $10^{-2}$. We also see that our method yields considerably lower errors at each $r$ compared to standard score matching. For a visual representation of the results, Figures 1b and 1c show a scatter plot of additional held-out samples from $\pi_X$ (which were not used during training) and these samples rotated into our discovered basis $\widetilde{U}$ when taking $d = 3$. In the learned basis, non-Gaussianity in the problem has been concentrated to the first two directions, and the third direction is now essentially independent of the first two.

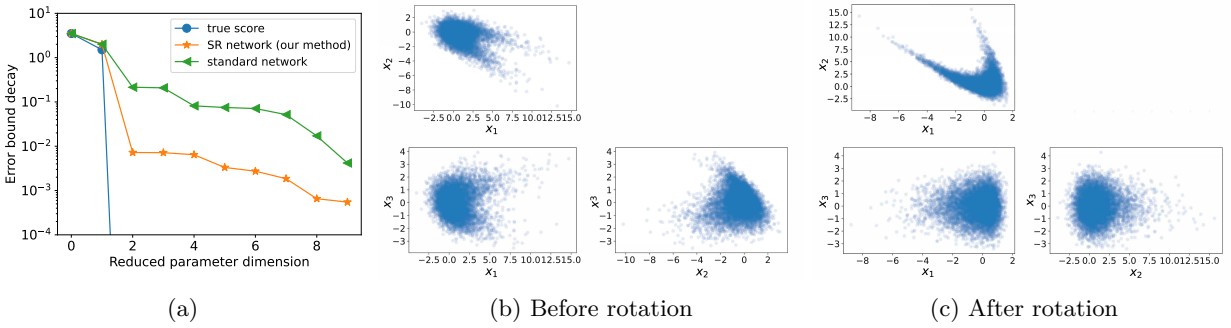

(a)        (b) Before rotation        (c) After rotation

Figure 1: Embedded banana problem: (a) ideal parameter reduction error bound, score matching error bound (standard network), and score ratio matching error bound (our method). Our method better captures the subspace where the target distribution $\pi_X$ deviates from the reference distribution. Middle and right: scatter plots of held-out samples from the embedded banana distribution with $d = 3$ (b) before (in the original random basis) and (c) after rotation by the learned basis $U$ for $X$. We observe that non-Gaussianity has been concentrated in the first two directions, and the third direction is now essentially independent of the first two.

### 6.2 PDE-constrained inverse problem

Next, we consider an inverse problem where the forward model involves an elliptic partial differential equation (PDE). The inference parameter describes the permeability field on a two-dimensional domain $\mathcal{D}$ and the observations are pointwise measures of the corresponding pressure field. This so-called "Darcy flow" problem is a widely-used test case in the literature on nonlinear Bayesian inverse problems (Stuart, 2010; Iglesias et al., 2014; Cui et al., 2014). The permeability field $e^\varrho$ and the pressure field $u$ are related by the following Poisson equation (Neuman & Yakowitz, 1979; McLaughlin & Townley, 1996; Carrera et al., 2005; Sun, 2013):

$$\begin{cases} \nabla \cdot (e^{\varrho} \nabla u) = 0, & \text{in } \mathcal{D} := [0,1]^2 \,, \\ u(\xi_1, 0) = 0 \quad u(\xi_1, 1) = 1 \\ \frac{\partial u}{\partial \boldsymbol{n}} = 0 \text{ for } \xi_2 \in \{0, 1\} \,. \end{cases} \tag{11}$$

Neumann boundary conditions on the left and right boundaries impose a zero-flux condition modeling impermeable layers on either side of an aquifer, and Dirichlet boundary conditions on the top and bottom boundaries correspond to fixed pressures.

We endow the log-permeability $\varrho$ with a Gaussian prior $\mathcal{N}(0, \Sigma)$ where $\Sigma$ is the Matérn covariance defined by the differential operator

$$C = (\delta \mathrm{I} - \gamma \Delta)^{-2}$$

where $\Delta$ is the Laplacian operator, and $\delta$ and $\gamma$ control the variance and correlation of the prior realizations (Lindgren et al., 2011).

The inference parameters $X$ are the leading $n = 100$ coefficients of the log-permeability in the Karhunen-Loève expansion of the Gaussian prior. Given that the solution operator mapping the permeability field to the pressure field, $e^{\varrho} \mapsto u$, is nonlinear, the inverse problem is nonlinear even without accounting for the exponential in the parameterization of the permeability. Let $\mathcal{F}$ denote the forward model mapping the parameter $x$ to $m = 100$ values of $u$ collected on a uniform $10 \times 10$ grid on $[0.1, \, 0.9]^2$. Observations for the parameter $x$ follow the model $Y = \mathcal{F}(x) + \epsilon$, where $\epsilon \sim \mathcal{N}(0, 10^{-3} I_n)$ and define the likelihood function $\pi_{Y|X}(\cdot | x)$. The resulting distribution of interest in this problem is the posterior $\pi_{X|Y}$. We take the prior covariance parameters to be $\delta = 0.5$ and $\gamma = 0.1$. Figure 2 shows three realizations of the log-permeability field drawn from the prior, the corresponding pressure fields, and observations.

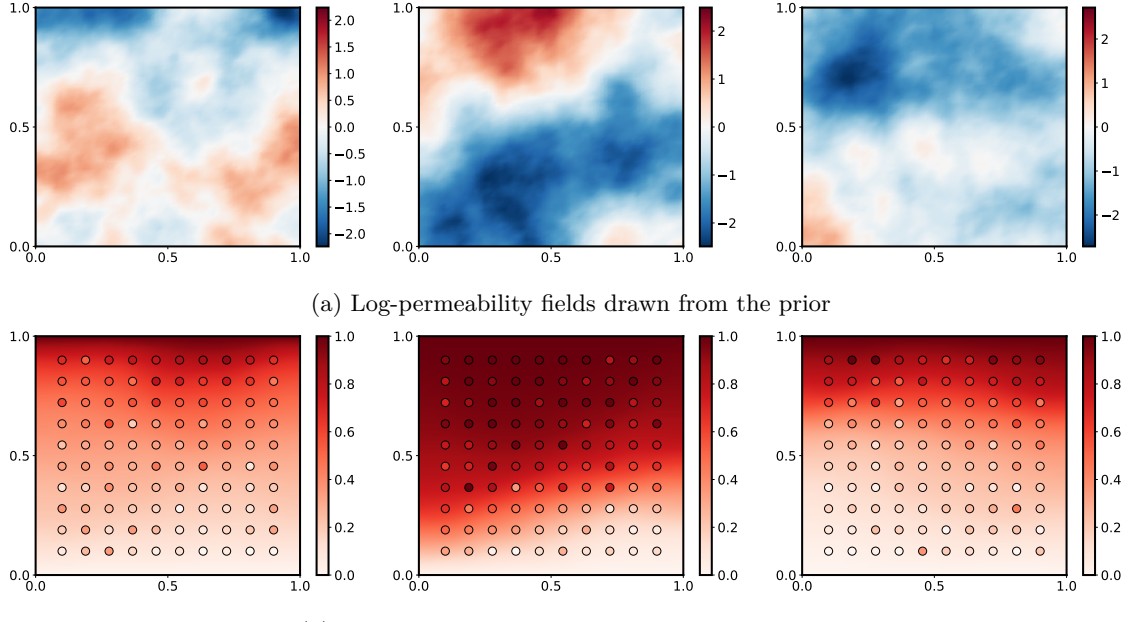

(a) Log-permeability fields drawn from the prior

(b) Corresponding pressure fields and observations

Figure 2: Darcy flow problem: example prior realizations of the log-permeability fields and corresponding pressure fields and observations

Here we fix the sample size to $N = 90000$ and apply both the single network (Algorithm 1) and iterative-deflated (Algorithm 2) versions of score ratio dimension reduction, taking the number of eigenvectors extracted at each deflation step to be $\ell = 1, 2,$ or $3$. Figures 3a and 3b compare the error bounds achieved by the optimal parameter and observation bases identified by our methods. We see that parameter reduction error achieved by the single score-ratio network separates from the optimal error earlier than the error achieved by the iterative deflated methods, and that overall $\ell = 1$ performs the best by a small margin. Each

of the networks performs similarly well for observation reduction. Figure 4 compares the four leading true and approximated parameter and observation basis vectors for the $\ell = 1$ iterative-deflated method, which visually match well (up to an immaterial change in sign).

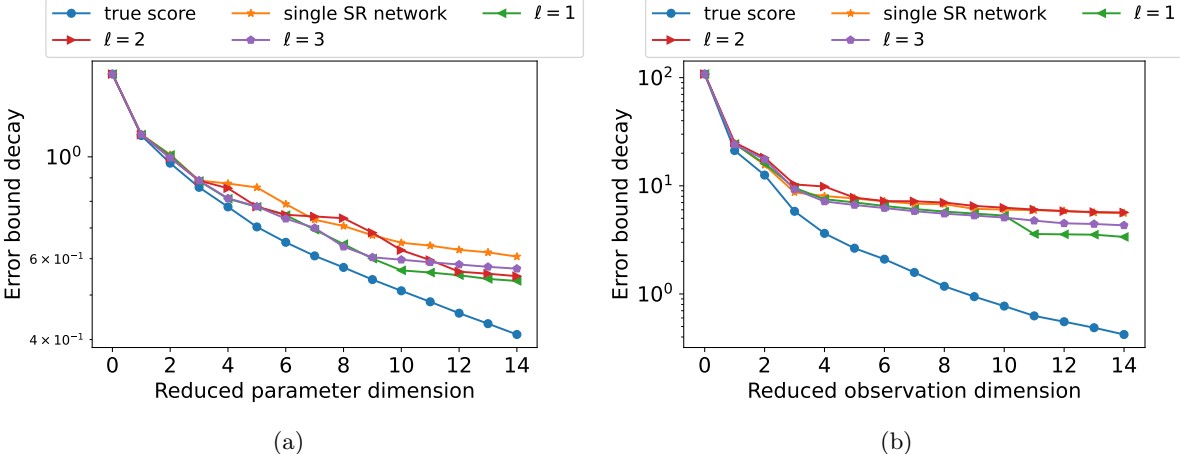

(a)                                                                (b)

Figure 3: Darcy flow problem: error bounds for parameter (a) and observation (b) subspaces identified by various methods. $\ell$ is the number of basis vectors extracted at each deflation step.

### 6.3  Flux, a quantity-of-interest inference example

In this example, we consider a simulation-based inference problem related to the Darcy flow problem of Section 6.2. The observation model is the same, and though the permeability remains uncertain, the parameter of interest is now the log of the flux across the lower boundary of the domain—a scalar-valued function of the permeability and the corresponding pressure. The log-flux is defined as

$$q = \log \int_{\Gamma_{\mathrm{B}}} e^\varrho \nabla u \cdot \mathbf{n} \, \mathrm{d}s \in \mathbb{R}, \tag{12}$$

where $\Gamma_{\mathrm{B}} = [0,1] \times \{0\}$ denotes the bottom boundary. For this problem, we introduce a source term $f$ on the right hand side of the Darcy flow PDE,

$$\begin{cases} \nabla \cdot (e^\varrho \nabla u) = f, & \text{in } \mathcal{D} \coloneqq [0,1]^2 \,, \\ u(\xi_1, 0) = 0 \quad u(\xi_1, 1) = 1 \\ \frac{\partial u}{\partial \boldsymbol{n}} = 0 \text{ for } \xi_2 \in \{0,1\} \,, \end{cases} \tag{13}$$

where the selected source term $f(\boldsymbol{\xi}) = 5 \exp\left(-20\|\boldsymbol{\xi} - \mathbf{c}\|^2\right)$, with $\mathbf{c} = (0.2, \ 0.2)$, models groundwater recharge.

The posterior distribution of interest in this problem is $\pi_{Q|Y}$, which follows from the joint distribution $\pi_{Q,Y}$. We note in this case that evaluating the unnormalized posterior density (and thus evaluating the score function) is not tractable; more details are given in Appendix B.3. We seek to reduce the dimension of the observations $Y$ via Algorithm 1 using $N = 10000$ samples. Figure 5a shows bounds on posterior approximation error that are estimated using our learned score-ratio network. The fast decay in this bound suggests that the posterior should be well approximated using relatively low-dimensional observation projections, e.g., $s = 4$. To validate the utility of the corresponding learned basis $\widetilde{V}_s$, we perform inference using a conditional normalizing flow (specifically an unconstrained monotonic neural network (UMNN) as described in Wehenkel & Louppe (2019)) trained from samples of $\pi_{Q,Y}$; see Appendix B.2 for the implementation details. We build two such flows: one depending on the full-dimensional observations and the other depending on the reduced observations of dimension $s = 4$ defined by the basis $\widetilde{V}_s$. We compare the performance of the UMNN flows

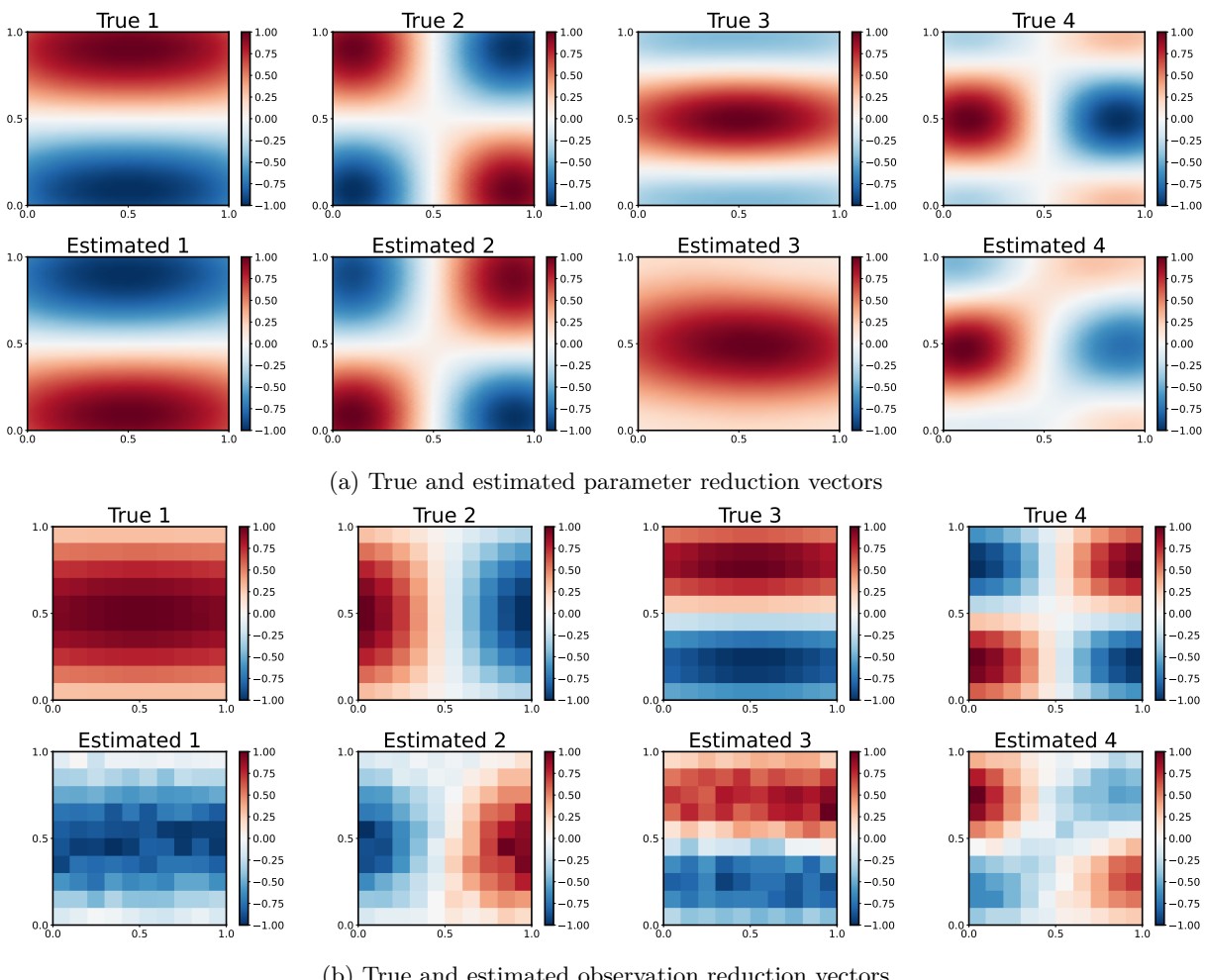

(a) True and estimated parameter reduction vectors

(b) True and estimated observation reduction vectors

Figure 4: Darcy flow problem: the four leading parameter and observation reduction vectors from the true diagnostic matrix and our score ratio estimates of them. We note that reduction vectors are equivalent up to a change in sign.

learned with an increasing number of map training samples (independent from the samples used to learn the score ratio approximation) to true posterior predictive samples computed via MCMC.[4]

Figure 6 shows posterior samples generated using MCMC and approximate posterior samples generated by the conditional normalizing flows with and without dimension reduction. Samples from the reduced-dimensional inference procedure more closely match the true posterior samples, across the full range of training set sizes. For quantitative comparisons, we let $\mathcal{Q}$ and $\widehat{\mathcal{Q}}$ denote sets of MCMC samples and flow-generated samples, respectively, and let $F$ and $\widehat{F}$ denote their respective empirical cumulative distribution functions. The Kolmogorov–Smirnov (KS) statistic is defined as $K(\mathcal{Q}, \widehat{\mathcal{Q}}) = \sup_q |F(q) - \widehat{F}(q)|$. Figure 5b shows the decay of the KS statistics as a function of the number of flow training samples, averaged across 10 different realizations of the observation variable $Y$. We see that reducing the observation dimension prior to performing inference yields lower KS statistics on average for any number of training samples.

---

[4]We perform MCMC on the high-dimensional latent parameters $X$ described in Section 6.2, and then compute log-flux samples $Q$ via (12).

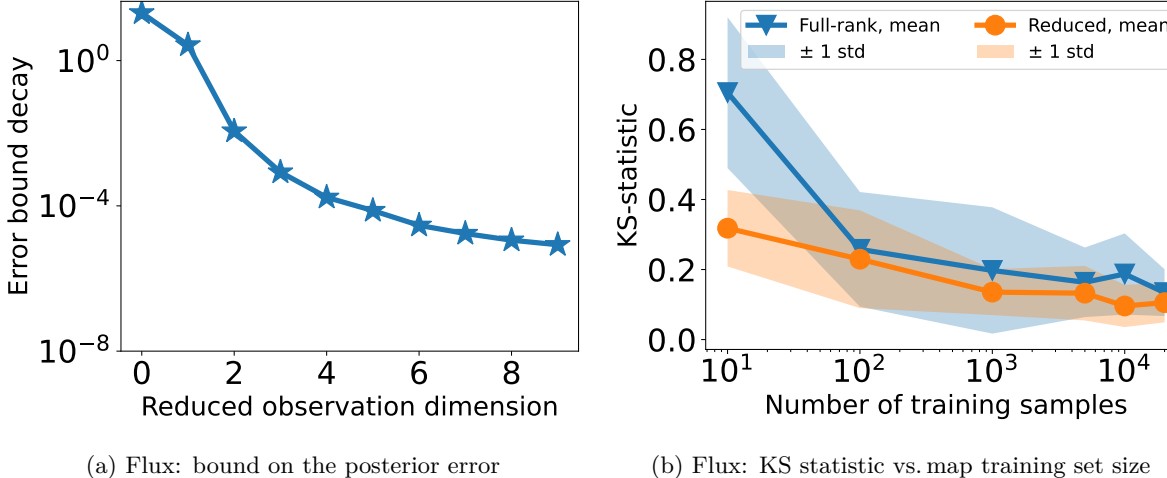

(a) Flux: bound on the posterior error

(b) Flux: KS statistic vs. map training set size

Figure 5: Flux problem: (a) decay of the estimated error bounds that may used to select the dimension of the reduction observation; (b) average KS statistic versus the number of training samples. Reducing the dimension of the observation enables higher-fidelity inference.

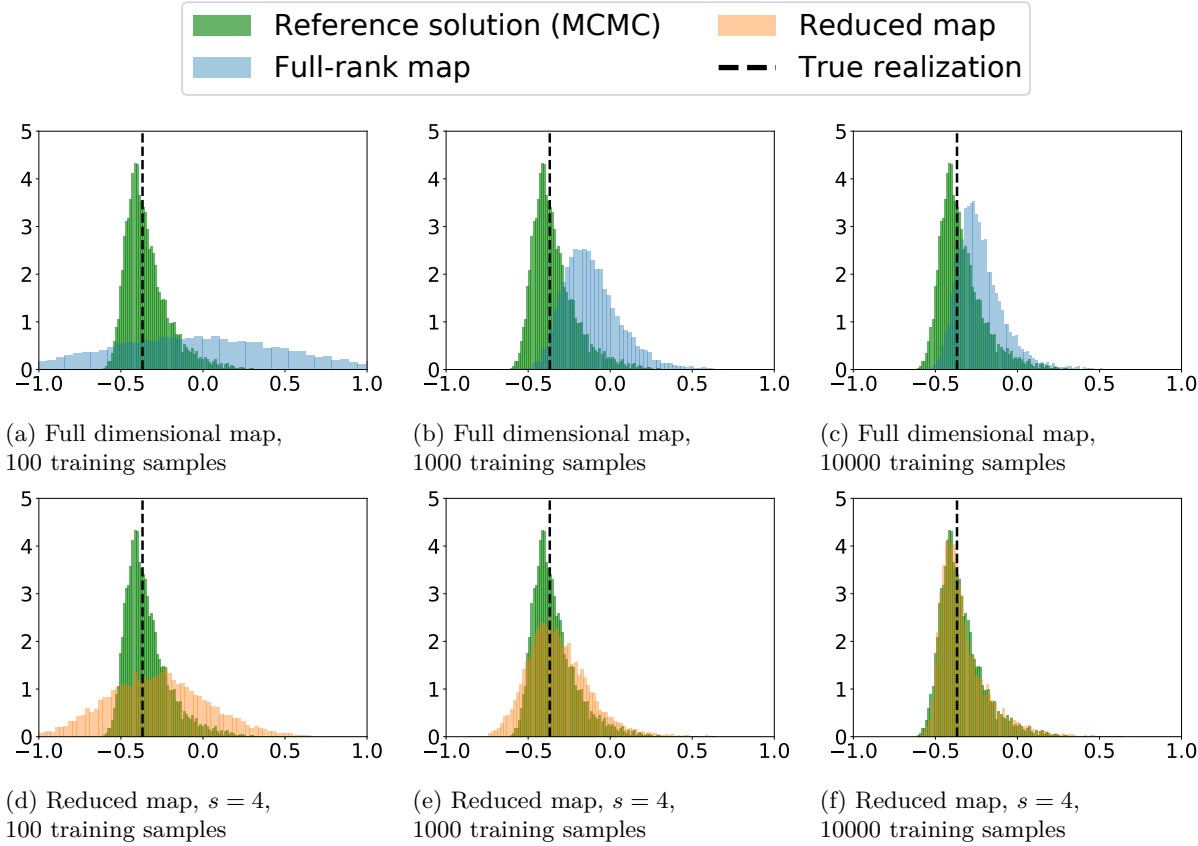

(a) Full dimensional map,
100 training samples

(b) Full dimensional map,
1000 training samples

(c) Full dimensional map,
10000 training samples

(d) Reduced map, $s = 4$,
100 training samples

(e) Reduced map, $s = 4$,
1000 training samples

(f) Reduced map, $s = 4$,
10000 training samples

Figure 6: Flux problem: comparison of samples from the full dimensional maps (without dimension reduction) and reduced maps (with dimension reduction) to a reference solution computed using MCMC for different amounts of map training data (left to right).

### 6.4 Energy price modeling

We now consider a conditional generative modeling problem related to energy market modeling. In the United States, there are seven independent system operators (ISOs) that operate competitive wholesale electricity markets where generators and resellers can buy and sell power. For renewable generators and microgrid operators, these markets provide important revenue opportunities that can improve the economic viability of new and existing projects. A generator's ability to capture these financial opportunities, however, is dependent on their ability to forecast wholesale electricity prices days ahead of time. Wholesale electricity prices vary by location and market and can be decomposed into the sum of three components: the energy price, the congestion price, and the cost of transmission losses. Energy prices are related to the cost of generation across an entire ISO and are constant over the entire region. Congestion costs vary by location and are nonzero when the physical constraints of electrical components, like transmission lines or transformers, are reached. As a step toward probabilistic forecasting, we investigate the conditional relationship between energy prices of the PJM ISO, which services much of the north-mid Atlantic United States, and temporal and weather-related covariates.

Our training dataset contains $N = 20000$ energy price and observation realizations from January 2020 through February 2023. Energy prices regularly take values across several orders of magnitude. As a pre-processing step, we take the natural log of the energy prices, then shift and scale the samples by its mean and standard deviation. Each observation realization comprises temperature and cloud coverage data from 43 weather stations around the serviced region (see Figure 8a), as well as a temporal encoding $\tau = (\sin(2\pi\text{DoY}/365), \cos(2\pi\text{DoY}/365), \sin(2\pi\text{HoD}/24), \cos(2\pi\text{HoD}/24))$, where DoY and HoD denote the day of the year and hour of the day of the prediction. This time encoding allows us to model the seasonal and daily patterns of the energy market. The parameter $X$ and observation $Y$ (i.e., temperature, cloud coverage and $\tau$) dimensions are $n = 1$ and $m = 90$, respectively.

We use this dataset to estimate the score ratio for $\pi_{X|Y}$ and thus the diagnostic matrix $H_{\text{CMI}}^Y$ via Algorithm 1. In Figure 7a, we observe a sharp drop in the estimated posterior error from observation dimension reduction, suggesting that the leading $s = 4$ modes should capture most of the information carried by the full observation about the parameter. Figures 8b and 8c plot the two leading basis vectors for the observation, and labels the observation components with the highest magnitudes. We see that the first vector primarily captures the temperature predictions at a few key cities. The second vector gives the most weight to the temporal encoding. Figure 9 plots the temperature and cloud coverage contribution of the leading four basis vectors on a map of the United States. We see that many vectors have interpretable structures. For example, the temperature component of $v_3$ seems to be a weighted average of the temperature predictions across the weather stations.

As before, we validate the learned reduction vectors by comparing the performance of approximate inference using full- and reduced-dimensional conditional normalizing flows as a function of the number of training samples from $\pi_{X,Y}$. While the ground-truth posterior distributions are not available in this problem, we do have the "true" realized value of the energy price. Thus we report the continuous ranked probability score (CRPS)[5], (Gneiting & Raftery, 2007) for the approximate posterior samples, a predictive metric used to evaluate the performance of probabilistic forecasts, on average over 100 held-out energy-price samples. Figure 7b shows that the reduced-dimensional flows are more predictive on average than the full-dimensional flow. We also note that the CRPS is stable with as few as 100 training samples for the reduced-dimensional map. Figure 10 shows six histograms of posterior samples generated from full- and reduced-dimensional flows trained with 10000 samples for different realizations of the observation variable. In two of the examples, we see that the reduced flows may capture underlying multi-modality in the posterior that the full-dimensional flow (i.e., without observation reduction) does not.

---

[5]The CRPS is defined as $\text{CRPS}(x^*, F) = \int (F(x) - \mathbb{1}_{x \geq x^*})\mathrm{d}x$ where $x^*$ is the true parameter value and $F$ is the cumulative distribution function of the approximate one-dimensional posterior distribution.

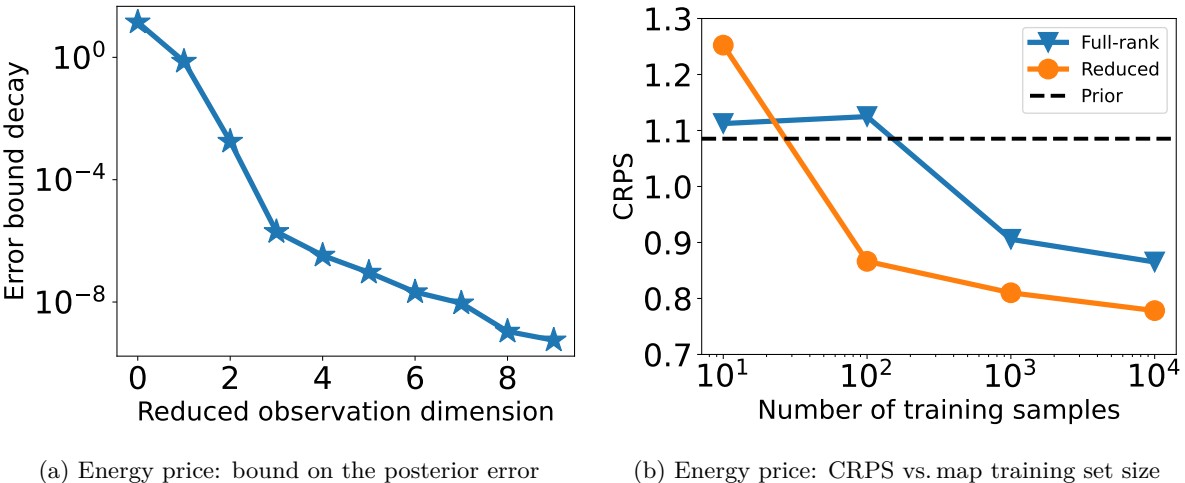

(a) Energy price: bound on the posterior error     (b) Energy price: CRPS vs. map training set size

Figure 7: Energy price problems: (a) decay of the estimated error bounds that may used to select the dimension of the reduction observation; (b) average CRPS versus the number of training samples. Reducing the dimension of the observation enables higher-fidelity inference.

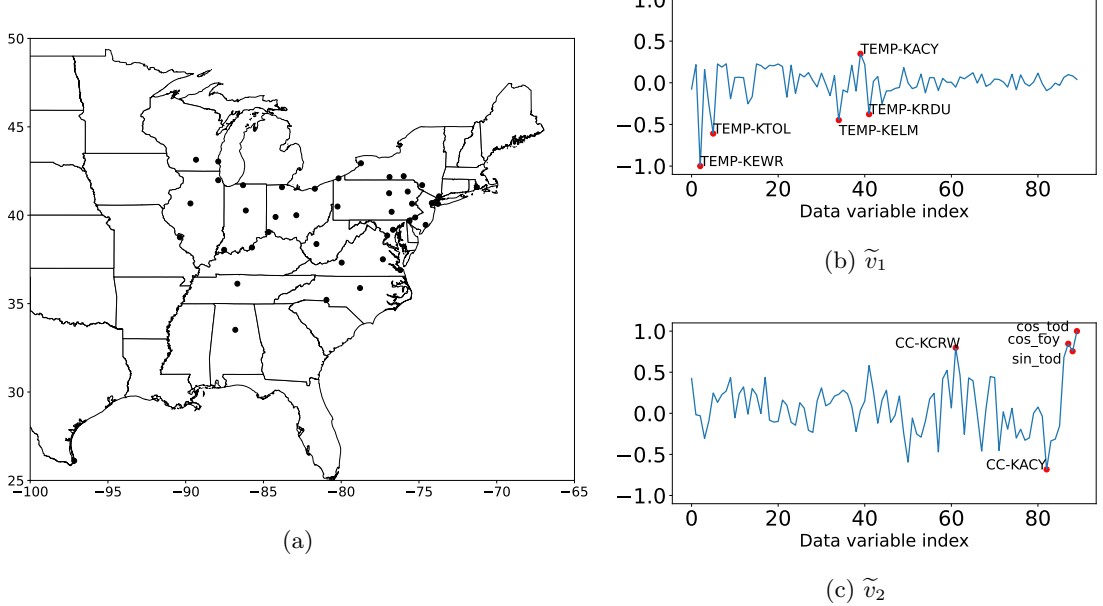

Figure 8: Energy price problem: (a) the location of weather stations, (b/c) the two leading reduced vectors plotted as line graphs with the five highest magnitude components labeled in red.

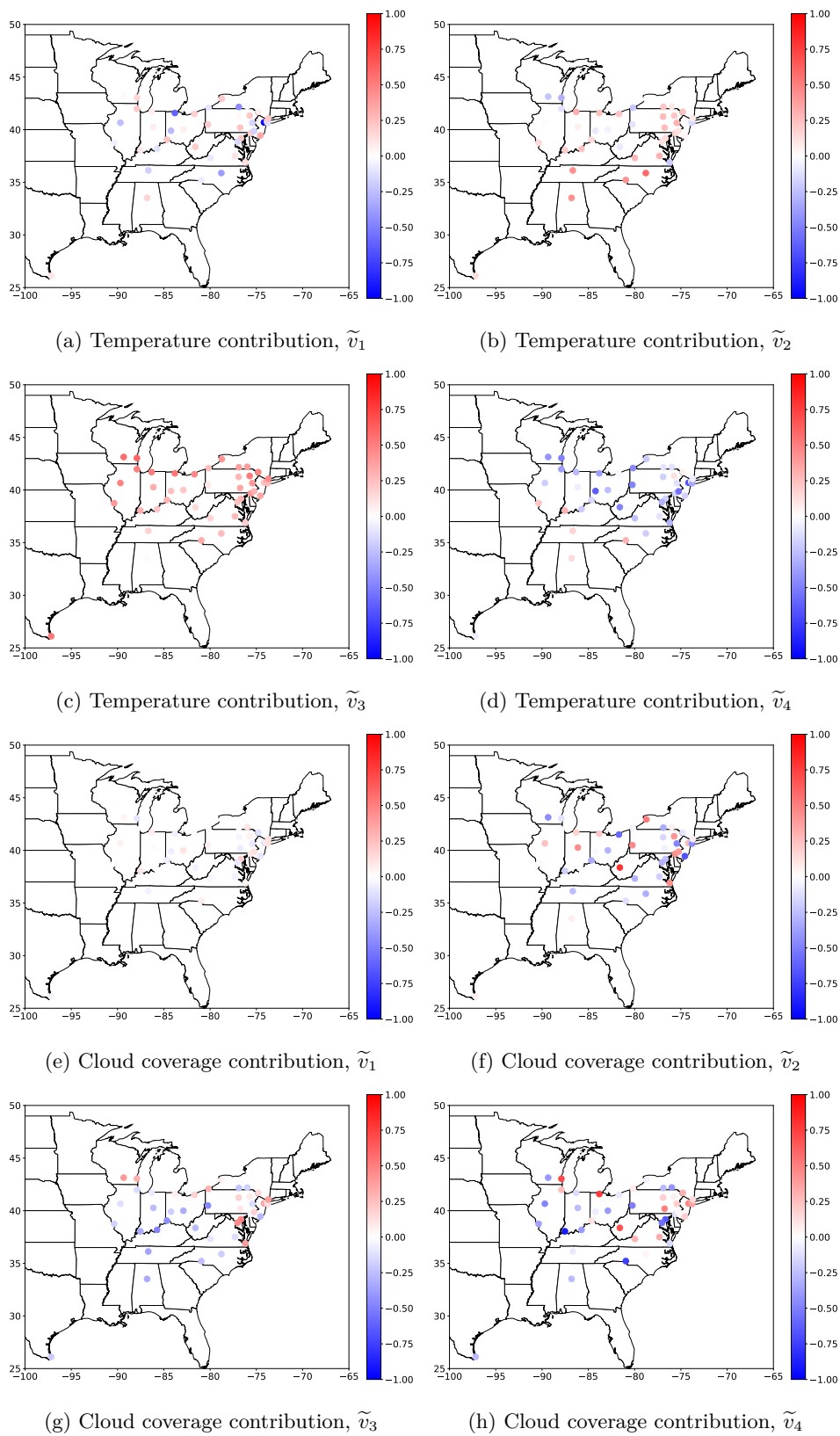

(a) Temperature contribution, $\widetilde{v}_1$      (b) Temperature contribution, $\widetilde{v}_2$

(c) Temperature contribution, $\widetilde{v}_3$      (d) Temperature contribution, $\widetilde{v}_4$

(e) Cloud coverage contribution, $\widetilde{v}_1$      (f) Cloud coverage contribution, $\widetilde{v}_2$

(g) Cloud coverage contribution, $\widetilde{v}_3$      (h) Cloud coverage contribution, $\widetilde{v}_4$

Figure 9: Energy price problem: the temperature and cloud coverage contributions of the leading four basis vectors for the observation space

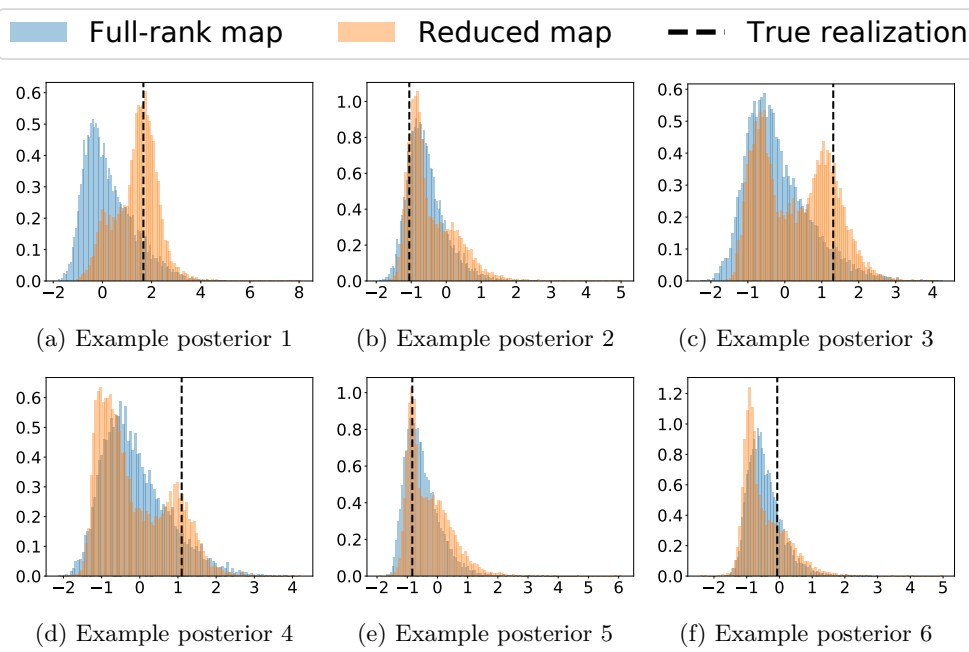

Figure 10: Energy price problem: six estimated posteriors (corresponding to different realizations of the observation) using both the full and reduced dimensional flows, trained with 10000 samples. The reduced flow samples seem more predictive of the true realization for each realized energy price. In (c,d), we observe that the reduced flow possibly captures bi-modality in the posterior.

# 7 Conclusions and future work

We introduce a framework for gradient-based dimension reduction of conditional distributions based on score ratio matching. In the Bayesian setting, for example, our methodology identifies low-dimensional subspaces of the parameter and observation spaces that best capture how the posterior departs from a chosen reference distribution and how the observations can be reduced. These subspaces are accurately identified from a score ratio network using a parameterization that exploits the score ratio's gradient structure, together with low-rank matrix regularizers and an eigenvector deflation technique. The discovered low-dimensional structure is shown to improve the accuracy of amortized inference and prediction in gradient-free settings for diverse applications, including PDE-constrained inverse problems and simulation-based inference of energy prices.

We outline several promising directions for future work. While the posterior approximation error guarantees in this work are currently limited to reduced *subspaces* of the parameters and observations, future work will investigate extensions to nonlinear dimension reduction; see Bigoni et al. (2022) for related work in a function approximation setting where gradients are available. In addition, it will be valuable to understand how the intrinsic dimension of the inference problem, along with the smoothness of the underlying densities, affect the number of samples $N$ needed to estimate the score ratio and to uncover this low-dimensional structure. For example, if one can characterize how the error in an estimated score ratio $\hat{w}_\theta^N$

$$\mathbb{E}_{\pi_{X,Y}} \left\| \hat{w}_\theta^N(x,y) - \nabla_x \log\left(\frac{\pi_{X|Y}(x|y)}{\rho(x)}\right) \right\|^2,$$

in expectation over the sample, scales with an appropriate intrinsic dimension and sample size, then Theorem 3.3 could be reformulated using user-specified parameters. We also note that Theorem 3.3 currently addresses parameter dimension reduction but not observation dimension reduction. Future work will aim to establish an analogous result for observation reduction, thereby mirroring the duality between Theorem 2.2 and Theorem 2.3.

**Acknowledgments**

The authors extend their gratitude to Olivier Zahm for his insights on the methodology and theoretical results in this work, and to Matt Parno for his help constructing the energy price problem and interpreting the results of our method. RB acknowledges support from the von Kármán instructorship at Caltech, the US Department of Energy AEOLUS center (award DE-SC0019303), the Air Force Office of Scientific Research MURI on "Machine Learning and Physics-Based Modeling and Simulation" (award FA9550-20-1-0358) and the Department of Defense (DoD) Vannevar Bush Faculty Fellowship (award N00014-22-1-2790) held by Andrew M. Stuart. MB and YM acknowledge support from the US Department of Energy, Office of Advanced Scientific Computing Research, M2dt MMICC center under award DE-SC0023187.

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
