# A Technical definitions and proofs

## A.1 Log-Sobolev inequalities

**Definition A.1** (Logarithmic Sobolev inequality). *A random variable $\Theta$ with density $\pi_\Theta$ on $\mathbb{R}^{d_\theta}$ satisfies a logarithmic Sobolev inequality (or log-Sobolev inequality) if there exists a constant $C < \infty$ such that*

$$\mathbb{E}\left[h(\theta)\log\left(\frac{h(\theta)}{\mathbb{E}[h(\theta)]}\right)\right] \leq \frac{C}{2}\mathbb{E}\left[\|\nabla h(\theta)\|^2 h(\theta)\right] \tag{14}$$

*holds for any smooth function $h : \mathbb{R}^{d_\theta} \to \mathbb{R}_{\geq 0}$. The smallest constant $C = C(\pi_\Theta)$ such that (14) holds is called the log-Sobolev constant of $\Theta$.*

**Definition A.2** (Subspace logarithmic Sobolev inequality). *A random variable $\Theta$ with density $\pi_\Theta$ on $\mathbb{R}^{d_\theta}$ satisfies the subspace logarithmic Sobolev inequality if there exists a constant $\overline{C} < \infty$ such that for any unitary matrix $W \in \mathbb{R}^{d_\theta \times d_\theta}$ and for any block decomposition $W = [W_t \ W_\perp]$ with $W_t \in \mathbb{R}^{d_\theta \times t}$, $t \leq d_\theta$, and for any $\theta_\perp \in \mathbb{R}^{d_\theta - t}$, the conditional random variable $\Theta_t | \Theta_\perp = \theta_\perp$ with $\Theta_t = W_t^\top \Theta$ and $\Theta_\perp = W_\perp^\top \Theta$ satisfies the log-Sobolev inequality with constant*

$$C(\pi_{\Theta_t | \Theta_\perp = \theta_\perp}) \leq \overline{C}. \tag{15}$$

*The smallest constant $\overline{C} = \overline{C}(\pi_\Theta)$ for which (15) holds for all conditionals is called the subspace log-Sobolev constant of $\Theta$.*

## A.2 Proof of Theorem 3.2

Let

$$J^*(\theta) = \frac{1}{2}\mathbb{E}_\pi\left\|w_\theta(x,y) - \nabla_x\log\left(\frac{\pi_{X|Y}(x|y)}{\rho(x)}\right)\right\|_2^2.$$

We expand the squared norm to obtain

$$J^*(s_\theta) = \frac{1}{2}\mathbb{E}_\pi\left[w_\theta(x,y)^\top w_\theta(x,y) + \nabla_x\log\left(\frac{\pi_{X|Y}(x|y)}{\rho(x)}\right)^\top \nabla_x\log\left(\frac{\pi_{X|Y}(x|y)}{\rho(x)}\right)\right.$$
$$\left. - 2w_\theta(x,y)^\top\nabla_x\log\left(\frac{\pi_{X|Y}(x|y)}{\rho(x)}\right)\right].$$

Note the second term, $\nabla_x\log\left(\frac{\pi_{X|Y}(x|y)}{\rho(x)}\right)^\top\nabla_x\log\left(\frac{\pi_{X|Y}(x|y)}{\rho(x)}\right)$, does not depend on the network parameters, and thus need not be included in our optimization objective. The following steps rewrites the third term into quantities we can evaluate:

$$\mathbb{E}_{\pi_{X,Y}}\left[w_\theta(x,y)^\top\nabla_x\log\left(\frac{\pi_{X|Y}(x|y)}{\rho(x)}\right)\right] = \mathbb{E}_{\pi_Y}\mathbb{E}_{\pi_{X|Y}}\left[w_\theta(x,y)^\top\nabla_x\left(\frac{\pi_{X|Y}(x|y)}{\rho(x)}\right)\frac{\rho(x)}{\pi_{X|Y}(x|y)}\right]$$

$$= \mathbb{E}_{\pi_Y}\int \pi_{X|Y}(x|y)w_\theta(x,y)^\top\nabla_x\left(\frac{\pi_{X|Y}(x|y)}{\rho(x)}\right)\frac{\rho(x)}{\pi_{X|Y}(x|y)}\,\mathrm{d}x$$

$$= \mathbb{E}_{\pi_Y}\int \rho(x)w_\theta(x,y)^\top\nabla_x\left(\frac{\pi_{X|Y}(x|y)}{\rho(x)}\right)\,\mathrm{d}x$$

$$= -\mathbb{E}_{\pi_Y}\int \mathrm{Tr}(\nabla_x(\rho(x)w_\theta(x,y)^\top))\left(\frac{\pi_{X|Y}(x|y)}{\rho(x)}\right)\,\mathrm{d}x$$

$$= -\mathbb{E}_{\pi_Y}\int \pi_{X|Y}(x|y)\left[\frac{\nabla_x\rho(x)^\top}{\rho(x)}w_\theta(x,y) + \mathrm{Tr}(\nabla_x w_\theta(x,y))\right]$$

$$= -\mathbb{E}_{\pi_Y}\mathbb{E}_{\pi_{X|Y}}\nabla_x\log\rho(x)^\top w_\theta(x,y) + \mathrm{Tr}(\nabla_x w_\theta(x,y))$$

$$= -\mathbb{E}_{\pi_{X,Y}}\nabla_x\log\rho(x)^\top w_\theta(x,y) + \mathrm{Tr}(\nabla_x w_\theta(x,y)).$$

The fourth line follows from integration by parts. This leads to the final result that

$$J^*(s_\theta) = \mathbb{E}_\pi \left[ \frac{1}{2} w_\theta(x,y)^\top w_\theta(x,y) + \text{Tr}(\nabla_x w_\theta(x,y)) + \nabla_x \log \rho(x)^\top w_\theta(x,y) \right]$$
$$+ \mathbb{E}_\pi \left[ \nabla_x \log \left( \frac{\pi_{X|Y}(x|y)}{\rho(x)} \right)^\top \nabla_x \log \left( \frac{\pi_{X|Y}(x|y)}{\rho(x)} \right) \right].$$

### A.3 Proof of Theorem 3.3

We start by stating a result achieved in the proof of Corollary 2.11 of Zahm et al. (2022). Let $U = [U_r \ U_\perp] \in \mathbb{R}^{n \times n}$ be a unitary matrix, and define $P_\perp = U_\perp U_\perp^\top$. Let $w(x,y) = \nabla_x \log \left( \pi_{X|Y}(x|y)/\rho(x) \right)$. Then

$$\mathbb{E} \left[ D_{\text{KL}}(\pi_{X|Y} || \widetilde{\pi}_{X|Y}) \right] \leq \frac{1}{2} \mathbb{E} \| P_\perp w \|^2.$$

Indeed it is from this result the diagnostic matrix is derived, noting that

$$\mathbb{E} \| P_\perp w \|^2 = \mathbb{E} \left[ \text{Tr}(P_\perp w w^\top P_\perp) \right] = P_\perp \underbrace{\mathbb{E} \left[ w w^\top \right]}_{H_{\text{CDR}}^X} P_\perp.$$

Let $w_\theta$ denote our approximation of the $w$. We have the following chain of inequalities:

$$\mathbb{E} \left[ D_{\text{KL}}(\pi_{X|Y} || \widetilde{\pi}_{X|Y}) \right] \leq \frac{1}{2} \mathbb{E} \| P_\perp w \|^2$$
$$= \frac{1}{2} \mathbb{E} \| P_\perp (w - w_\theta + w_\theta) \|^2$$
$$\leq \frac{1}{2} \left( 2\mathbb{E} \| P_\perp (w - w_\theta) \|^2 + 2\mathbb{E} \| P_\perp w_\theta \|^2 \right)$$
$$\leq \mathbb{E} \| w - w_\theta \|^2 + \mathbb{E} \| P_\perp w_\theta \|^2,$$

where the third step follows from the triangle inequality, and the fourth step follows given $P_\perp$ is an orthogonal projector. We assume $\mathbb{E} \| w - w_\theta \|^2 < \epsilon$. Taking $U$ to be the eigenvectors of the estimated diagnostic matrix $\widehat{H}_{\text{CDR}}^X = \mathbb{E} \left[ w_\theta w_\theta^\top \right]$, yields $\mathbb{E} \| P_\perp w_\theta \|^2 = \sum_{k>r} \lambda_k$, completing the result.

### A.4 Proof of Theorem 5.1

*Proof.* This can be seen by considering the eigenvalue expansion of $A$

$$A = \sum_{i=1}^d \lambda_i \phi_r \phi_r^\top.$$

Then

$$\Phi_r \Phi_r^\top A = \sum_{i=1}^d \lambda_i (\Phi_r \Phi_r^\top) \phi_r \phi_r^\top = \sum_{i=1}^d \lambda_i \Phi_r (\Phi_r^\top \phi_r) \phi_r^\top = \sum_{i=1}^r \lambda_i \phi_r \phi_r^\top$$

given the orthogonality of the eigenvectors. Therefore

$$PA = A - \sum_{i=1}^r \lambda_i \phi_r \phi_r^\top = \sum_{i=r+1}^d \lambda_i \phi_r \phi_r^\top$$

and $PAP = \sum_{i=r+1}^d \lambda_i \phi_r \phi_r^\top$ follows similarly.

$\square$

### A.5 Proof of Theorem 5.2

We first note that

$$\nabla_y w_P(x,y) = P^X \nabla_y \nabla_x h(x, P^Y y) P^Y$$

where $h(x,y) = \nabla_y \nabla_x \log \pi_{X,Y}(x,y)$. Substituting $w_P$ into the definitions of the diagnostic matrices yields

$$\widetilde{H}_{\mathrm{CDR}} = P^X \underbrace{\mathbb{E}_{\pi_{X,Y}} \left[ \nabla_x \log \left( \frac{\pi_{X|Y}(x|P^Y y)}{\rho(x)} \right) \nabla_x \log \left( \frac{\pi_{X|Y}(x|P^Y y)}{\rho(x)} \right)^\top \right]}_{H} P^X$$

$$\widetilde{H}_{\mathrm{CMI}}^Y = P^Y \underbrace{\mathbb{E}_{\pi_{X,Y}} \left[ \nabla_y \nabla_x h(x, P^Y y)^\top P^X P^X \nabla_y \nabla_x h(x, P^Y y) \right]}_{H'} P^Y.$$

## B   Additional details for the numerical results

### B.1  Score ratio network hyperparameters

For each problem we use the Adam (Kingma & Ba, 2015) optimizer with a learning rate $5 \times 10^{-3}$ and batch size 1000 to train the network. We take the nuclear norm regularization parameters to be $\lambda_1 = 1/n$ and $\lambda_2 = 1/m$, where $n$ and $m$ are the parameter and observation dimensions. As discussed in Song et al. (2020), directly evaluating the trace operator in the objective function $F$ is prohibitively expensive for even moderate dimensions $n$, and so we also make use of the sliced-score matching method proposed in that work with 100 projections. The network $\psi_\theta$ were taking to be fully connected. Table 1 summarizes all hyperparameter choices for the score ratio network. These parameters include the network depth (i.e., number of hidden layers), width (i.e., layer dimension), the training set size and number of epochs.

| Problem | $r', s'$ | Hidden layers | Layer dimension | Training set size | Epochs |
|---|---|---|---|---|---|
| Embedded banana | 10, NA | 1 | 32 | 1K | 5 |
| Darcy (single network) | 30, 30 | 3 | 30 | 90$K$ | 3 |
| Darcy ($\ell = 1, 2, 3$) | 10, 10 | 3 | 30 | 90$K$ | 3 |
| Flux | 1, 100 | 3 | 200 | 10K | 100 |
| Energy market | 1, 90 | 3 | 128 | 20K | 30 |

Table 1: Summary of score network hyperparameter choices

### B.2  Approximate inference via measure transport

Our underlying motivation for reducing the parameter and observation dimensions is to improve the performance of approximate sampling methods. One sampling strategy that has recently grown in popularity is parametric measure transport (which encompasses normalizing flows), where one attempts to couple the target distribution $\pi$ (e.g., the posterior distribution of interest) with a tractable reference distribution $\rho$ (in our case the standard Gaussian distribution) using a map $S$ so that $S_\sharp \rho = \pi$. Having access to this map permits generating i.i.d. samples from the target distribution by generating reference samples $z^{(n)} \sim \rho$ and evaluating the map at these samples $S^{-1}(z^{(n)}) \sim \pi$. We direct an interested reader to Marzouk et al. (2016); Papamakarios et al. (2021) and references therein for a comprehensive review of such methods.

In general, the accuracy of the generated posterior samples depends on (1) the expressivity of the transport map class, and (2) the ability to optimize the transport map. In practice, it is often seen that large training sets and computational effort are required to optimize expressive maps for high dimensional problems. Brennan et al. (2020); Baptista et al. (2022) show that dimension reduction can improve the fidelity of approximate posteriors, especially in the data- and compute-limited regimes.

In the numerical experiments of Sections 6.3 and 6.4, we utilize unconstrained monotonic neural networks (UMNNs) (Wehenkel & Louppe, 2019) implemented within the `nflows` software package (Durkan et al.,

2020) as the underlying transport map class. The map $S \colon \mathbb{R}^n \times \mathbb{R}^{\dim(y)} \to \mathbb{R}^n$ for sampling conditional distributions depends on the conditioning variable $y$. For an $(n = 1)$-dimensional parameter, the map takes the form

$$S(y, x) = \int_0^x f(y, t; \phi) \mathrm{d}t + \beta,$$

where $f \colon \mathbb{R}^{\dim(y)} \times \mathbb{R} \times \mathbb{R}^{\dim(\phi)} \to \mathbb{R}_{>0}$ is a strictly positive parametric function and $\beta \in \mathbb{R}$ is a scalar bias term. The function $f$ can be made arbitrarily complex using an unconstrained neural network whose output is forced to be strictly positive through a shifted ELU activation unit. Here $\phi$ denotes the parameters of this neural network, meaning the map parameters we train are $(\phi, \beta)$. For both example, the underlying networks have 5 hidden layers. We let the number of hidden features scale with the input-dimension, taking it to be equal to $\lceil 1.5 \dim(y) \rceil$. We train the map using Adam (Kingma & Ba, 2015) using a step-size of $5 \times 10^{-4}$. 10% of the training samples were held out to be a validation set and training was stopped when the objective evaluated on this validation set plateaus in value.

### B.3 Derivation of the posterior and score function for the Flux problem

The flux is defined as

$$q = \log \int_{\Gamma_{\mathrm{B}}} e^{\varrho} \nabla u \cdot \mathbf{n} \, \mathrm{d}s \in \mathbb{R}, \tag{16}$$

where $\Gamma_{\mathrm{B}} = [0, 1] \times \{0\}$ denotes the bottom boundary of the domain. As done section 6.2, we express the log-diffusivity field $\varrho$ in the Karhunen-Loève expansion of the Gaussian prior with coefficients $x$, defining the mapping $\mathcal{Q} \colon \mathbb{R}^{100} \to \mathbb{R}$

$$q = \mathcal{Q}(x).$$

The target posterior for this problem, $\pi_{Q|Y}$, can then be written as the following integral via change of measure formula

$$\pi_{Q|Y}(q|y) = \int_{\mathcal{Q}^{-1}(q)} \frac{\pi_{X|Y}(x|y)}{\|\nabla_x \mathcal{Q}(x)\|} \mathrm{d}x.$$

We note here that evaluating the unnormalized posterior and thus the posterior score function is intractable, as it requires the computing the integral over the pre-image of the mapping $\mathcal{Q}$.