# OpenReview forum: "Dimension reduction via score ratio matching"
_TMLR — Accepted by TMLR_

### Review · Reviewer_XpJ3 · 2024-11-20

**Summary Of Contributions:**

The paper introduces a novel framework for dimension reduction in scenarios without gradient information, addressing limitations of traditional gradient-based methods. It derives an objective from score-matching to learn score ratio functions, enabling computation of diagnostic matrices without gradients.

**Audience:**

Yes

**Claims And Evidence:**

Yes

**Requested Changes:**

Comparison with other dimension reduction techniques, such as t-SNE

Discussion about scalability and computation overheads

More real world demonstrations

**Strengths And Weaknesses:**

### **Strengths**
The framework extends dimension reduction to gradient-free settings, addressing a critical gap in existing methodologies.
It also has applicability to diverse areas, including PDE-constrained Bayesian inference and conditional generative modeling.

Regularization techniques effectively exploit the presumed low-dimensional structure of data.
The eigenvalue deflation algorithm enhances the identification of low-dimensional projections with limited data.

Experiments demonstrate the method’s superiority over standard score-matching techniques in relevant scenarios.

### **Weaknesses**
The score ratio learning and iterative deflation process may be computationally intensive for large-scale problems.

 While the experiments are thorough, additional real-world applications could better demonstrate practical impact.

The choice of regularization parameters might heavily influence performance, requiring careful tuning.

The method’s scalability to very high-dimensional or extremely large datasets remains unclear.

---

> ### Author Response · Authors · 2025-03-05
> **Reviewer XpJ3 response**
>
> Weaknesses
>
> * Addressed below
> * Addressed below
> * We agree there are certainly problems where the regularization parameters will need to be tuned for best results. We found that scaling them with the problem dimensions (taking lambda_1 = 1/n and lambda_2 = 1/m) worked reasonably well; we didn't attempt to tune them in order to present the fairest assessment of performance on a new problem where the optimal parameters are not known in advance.
> * Addressed below
>
> Requested Changes:
>
> * We have added a paragraph comparing our proposed method to other dimension reduction techniques, including PCA, UMAP, and t-SNE at the end of Section 3.
> * We have added a discussion of the computational costs and scalability of our methods in a paragraph at end of Section 5.
> * We, respectfully, believe that our numerical examples demonstrate the utility of our approach to many scientific domains. We have constructed a first idealized problem that exhibits the exact type of low-dimensional structure our method identifies, followed by three more challenging problems to demonstrate that our approach finds useful structure in real-world applications. In particular, the problems involving PDE-based forward models arise in many science and engineering applications (e.g., the geosciences). Reducing the dimension of the (possibly infinite-dimensional) parameters in these systems is essential to make Bayesian inference feasible with these forward models. The last problem on energy forecasting was based on real data provided by an industrial partner.

---

### Review · Reviewer_Sz6n · 2024-12-05

**Summary Of Contributions:**

This paper explores two types of low-dimensional structures in conditional distributions, such as posterior distributions in Bayesian settings. The first type involves approximating the target distribution as a low-dimensional update of a reference distribution, while the second type replaces conditioning variables with low-dimensional projections or summaries. Detecting these structures is achieved by constructing specific _diagnostic matrices_ that capture information from the gradient or Hessian.

The paper extends these methods to scenarios where gradients of distributions are unavailable, employing a score-matching procedure to approximate the gradient of a log-density. Specifically, using a neural network, the authors focus on approximating the gradient of the logarithm of the ratio between the target conditional distribution and a reference standard Gaussian distribution. This shift in focus to the score-ratio function is motivated by the ability to express diagnostic matrices in its terms. An iterative deflation method is also introduced to reduce the computational cost of constructing these diagnostic matrices.

The proposed methodology is evaluated on various datasets, including both real and simulated examples, to demonstrate its applicability and effectiveness.

**Audience:**

Yes

**Broader Impact Concerns:**

No one detected.

**Claims And Evidence:**

Yes

**Requested Changes:**

1. Address the weaknesses outlined above.
2. Explicitly refer to the Supplementary Material in the text where appropriate (e.g., Section 6: "Table 1 _in the Supplementary Material_ summarizes all network and training hyperparameter choices for each numerical example").
3. Use a different letter for variables in Lemma 5.1 to avoid confusion with the network \( w(x, y) \).

4. I list some minor inconsistencies and typos:
  - Section 2.2: The sentence "contribution to the latter sections" seems extraneous and should be removed.
  - Figure 1: Ensure consistency between figure labels and text. In the figure, the axes are labeled \( X_0, X_1, X_2 \), while the text uses \( X_1, X_2, X_3 \).
  - Double-check the bibliography: Address duplications (e.g., Cui et al., "Likelihood-informed dimension reduction for nonlinear inverse problems" is listed twice).
  - Equation (8): I assume the 1/N multiplies the entire expression, as MC is applied to approximate Equation (7) - use parenthesis to avoid confusion.

**Strengths And Weaknesses:**

**Strengths:**
The paper is generally well-written and clear, with only a few minor typographical errors (listed below). The proposed method is interesting, and its application to both simulated and real datasets effectively showcases its utility.

**Weaknesses:**
- It is unclear how sensitive the method is to certain factors, such as:
    - The robustness of results to changes in the neural network training dataset. How robust are the projections when multiple training datasets are considered?
    - The choice of the reference distribution. What happens if a different reference is considered?
    - The number of training samples used. While Figure 5 addresses sample size to some extent, the authors could expand on its implications. Under what conditions is a large sample size essential for the method to work well?
- If I understand correctly, the method performs linear projections through the eigendecomposition of diagnostic matrices; comparing its results with those from standard dimensionality reduction techniques like PCA, T-SNE, or UMAP would be helpful. Highlighting strengths and weaknesses relative to these methods would clarify the method's applicability.
- Could the author elaborate more on the computational cost implied by their method?

---

> ### Author Response · Authors · 2025-03-05
> **Reviewer Sz6n response**
>
> Weaknesses:
>
>
> * The stability of the learned score ratio networks and resulting projections to different training sets is an interesting and important aspect of the problem. In the problems we considered, we observed that the learned basis vectors were qualitatively similar across multiple training datasets. We have added a comment in the conclusions section on rigorously/theoretically assessing the statistical variability of the learned basis vectors as a promising avenue for future work.
>
>
>
> * We have added several remarks on the choice of reference distribution at the end of section 3, as part of a general discussion of properties of our reduction methods.
>
>
>
> * We have added comments on the necessary sample sizes in the conclusions section as well as plans to investigate this rigorously as future work.
>
>
>
> * We have added a paragraph comparing our proposed method to other dimension reduction techniques, including PCA, UMAP, and t-SNE at the end of Section 3.
>
>
>
> * We have added a discussion of the computationally costs and scalability of our methods in a paragraph at end of Section 5.
>
> Requested Changes:
>
> * We have added "in the Supplementary Material" where necessary.
> * We changed the basis vectors and matrix to $\phi, \Phi$ in Lemma 5.1
> * We have fixed each of these typos and inconsistencies; thank you for pointing these out!

---

### Review · Reviewer_n4op · 2025-02-11

**Summary Of Contributions:**

This paper studies the application of gradient-based dimension reduction, in settings where the gradient is not (directly) available. The proposed method approximates a posterior $\pi_{X|Y}$, by reducing the dimension of $(X, Y)$ using the transformations $(U, V)$. The proposed method proceeds in three steps: (i) approximate a score ratio, (ii) approximate diagnostic matrices, and (iii) estimate $U$ and $V$ using the leading eigenvectors of the estimated diagnostic matrices.

The authors bound the error of the approximate posterior, based on the error in the score ratio approximation and the rank of the transformation $U$. As far as I can see, there is no analogous result for the transformation $V$.

Next, the authors refine their method and propose learning a neural network for the score ratio, by taking advantage of the assumed low-dimensional structure of the posterior. Finally, the authors propose using an eigenvalue deflation method to iteratively learn a score-ratio network and corresponding columns for $U$ and $V$.

The paper examines the proposed algorithms on three examples: a multivariate distribution with low-dimensional non-Gaussianity, an inverse PDE problem, and energy pricing problem.

**Audience:**

Yes

**Broader Impact Concerns:**

No concerns.

**Claims And Evidence:**

Yes

**Requested Changes:**

See the above discussion on weaknesses.

I also have a few minor comments and questions:
* In Section 3, why is it "naive" to approximate $\nabla_x \log \pi_{X|Y}(x|y)$ and then compute the score ratio?
* In Theorem 3.3, the R.H.S on the inequality, $\lambda_k$ can be evaluated but, as far as I can tell, $\varepsilon$ cannot. This prevents us from computing the upper-bound in practice and this limitation should be acknowledged.
* How does the procedure in Section 4 change the error in the score-function network?
* In Section 6.2, the model is $Y = \mathcal F(x) + \varepsilon$ and $\varepsilon \sim \mathcal (0, 10^{-3} I)$. I'm curious: how is the $10^{-3}$ scaling determined?
* In Section 6.4, why are the authors believe there are $s = 4$ leading modes (say as opposed to $s = 3$ or $s = 5$)?
* I recommend splitting Figure 6 into two figures, and trying to order the figures based on when they are mentioned in the text.
* Can the authors report the formula for the continuous ranked probability score?

**Strengths And Weaknesses:**

Strengths:
* The paper proposes an approach to generalize score-based dimension reduction methods to cases where the gradient is not available in an analytical, or at least straightforward, form. To my knowledge, this is new.
* The paper offers an enlightening theoretical analysis. In particular, Theorem 3.3 provides an upper-bound on the expected KL-divergence, both in terms of the error introduced by the score network and the low-dimensional representation.
* The proposed algorithm is clear and not overly complicated. As far as I can tell, there are not too many tuning parameters to deal with. The main choices are: $\varepsilon_X$, $\varepsilon_Y$, architecture of score network. I find it very compelling that the ranks of $U$ and $V$ are not set manually, but computed to match the tolerances $\varepsilon_X, \varepsilon_Y$.
* The numerical experiments are diverse and analyzed in a good level of details. I appreciated that the authors examin how the performance of their algorithms changes with the size of the training set. Overall, I found these extended "case studies" quite insightful.
* Another strength of the method is that it provides an estimable error bound, which can be computed (as in Figures 6(a) and 6(b)).
* Identifying projections of $X$ and $Y$ is insightful, as showcased in Section 6.4.
* The paper is very well written.

Weaknesses:
* The error bounds are in terms of $\mathbb E_{\pi_Y} [D_{KL}]$, which is a reasonable choice, though other measures of disagreements between the approximation and the target may be consider. For example total variation and Wasserstein distances, which provide bounds on errors in the estimation of expectation values and quantiles. I recommend that the authors justify their choice of error measure (in terms of theoretical guarantees and mathematical convenience).
* Theorem 3.3 seems incomplete: can the authors add an inequality which accounts for the low-rank of $V$ (thereby mirroring the duality between Propositions 2.2 and 2.3), and another inequality which accounts for $\varepsilon$, $V$, and $U$ simultaneously? If this is not straightforward, I recommend adding a sentence or two as to why, and an acknowledgment that this could be future work.
* The steps in Sections 4 and 5 are interesting, and make sense but, as far as I can tell, the paper does not evaluate the benefits in terms of computation cost and the error potentially introduced in Section 4.
* In Figure 3, there does not seem to be a clear a difference between the trajectories (except for the "true score" benchmark). The results seem noisy. Can the authors add error bars or error shades? From Algorithm 1, it seems the main source of variation would be the data drawn from $\pi_{X,Y}$, so maybe the experiment could be repeated with different realizations of the data.
* In Section 6.4, the authors remark that the reduced map finds multiple modes, compared to the full-rank map. Why is this better? How do the authors know whether the modes are spurious or not, and is there a way to interpret these modes?

* The examples in Sections 6.2 and 6.3 admit a tractable score. For the example in Section 6.4, the authors do not provide enough details for me to assess whether it is straightforward to evaluate the score. If this falls within an SBI framework, this should be more clearly emphasized.

---

> ### Author Response · Authors · 2025-03-05
> **Reviewer n4op response**
>
> Weaknesses:
>
>
>
> * We agree with the reviewer that developing similar dimension reduction approaches that minimizer the posterior approximation error under different error metrics may be useful under different contexts (e.g., minimizing Wasserstein distances may be particularly useful when the distributions have atoms). The Kullback-Leibler divergence is natural for score-ratio approximations because of its links to the underlying log-Sobolev bounds on posterior approximation error, and also because of our stability result (Theorem 3.3) linking posterior error to the score approximation error.  The LSI bounds have been extended to \alpha-divergences in [Li et al. 2024, Bernoulli, DOI: 10.3150/23-BEJ1702], but not Wasserstein or TV. We have added a remark at end of section 3 to discuss alternative error metrics for dimension reduction methods.
>
>
>
> * Thank you for this comment. We were not immediately able to extend Theorem 3.3 to include the observation reduction. We have added a note in the conclusion that future work will seek an analogous result considering observation reduction.
>
>
>
> * Addressed below
>
>
>
> * We thank the reviewer for allowing us to clarify this point. The error bounds at r = 2 (the true underlying dimension) for the score network and our score ratio method are 0.2 and 0.01 respectively, a fairly clear difference. Given that this problem is a toy example meant to showcase the particular structure we are searching for, we believe this result is sufficient.
>
>
>
> * We believe this is a fair point. The improved CRPS values of the reduced maps give us some evidence that these modes are not spurious, as it shows they are generally more predictive. Also, while this is certainly not a guarantee of correctness, the new modes found via inference with reduced maps lie closer to the true parameter value. For this particular example, the parameter is a scaling of the log-energy price (the average price of a MW in the PJM market). This second mode can be interpreted as capturing the possibility of a high (but realistic) price for that date and time. This draws from particular intuition for this problem, which we felt was out of the scope of this paper and would distract from its main contributions.
>
>
>
> * We have added a small section in the appendix explaining why the score function is not tractable for the problem described in 6.4.
>
> Requested Changes:
>
>
> * By naive, we mean 'a natural first choice, but less effective than what we propose later.' Naive in that it does not exploit possible low dimensional structure that we propose in our paper.
>
>
>
> * We have added a statement directly following the theorem to acknowledge that eps is typically unknown in practice.
>
>
>
> * It is challenging to make broad claims, but our intuition is that both the regularization and network parameterization guide the network toward the posited low-dimensional structure we aim to capture. The parameterization effectively narrows the approximation space to functions that behave like gradients of ridge functions. The numerical example of Section 6.1 shows the combined benefits of learning the score ratio (vs the score function) and of our parameterization and regularization.
>
>
>
> * In practice, the scaling of the observation noise is determined by characterizing the precision of the measurement process. Papers considering this PDE governed inference problem typically choose the scaling between 10^{-4} -- 10^{-2}. See:
>     * Iglesias, Marco A., Kui Lin, and Andrew M. Stuart. "Well-posed Bayesian geometric inverse problems arising in subsurface flow." Inverse problems
>     * Cui, Tiangang, and Xin T. Tong. "A unified performance analysis of likelihood-informed subspace methods." Bernoulli,
>
>
> * We think one could take s = 3, 4, 5 as reasonable choices based on the fast decay of the error bound plotted for this problem.
>
>
>
> * We agree and have made this change to the figures.
>
>
>
> * We added the formula for CRPS in a footnote in section 6.4.

---

### Author Response · Authors · 2025-03-05
**General comment**

Thank you all for taking the time to construct great reviews. We are appreciative of the overall positive view of our submission. We stress that our interest is to propose a dimension reduction strategy that  targets a form of low-dimensional structure in Bayesian inference problems that is not directly exploited by alternative strategies such as PCA and t-SNE. Our approach is based on rigorous guarantees of the approximation error and it can be implemented in practice in the setting where gradients are unavailable. In the revised manuscript, we discuss each of the limitations raised by the reviewers. We have highlighted additions to the text in red.

---

### Author Response · Authors · 2025-04-22
**Thanks! Camera ready version has been uploaded**

We are delighted that our work has been accepted! Thank you to the reviewers and the area chair for the thoughtful feedback and for facilitating the review process. We have uploaded the camera-ready version of the paper.

---

### Decision · Action_Editor_bbpB · 2025-03-31

**Recommendation:** Accept with minor revision

**Comment:**

All reviewers agreed that the paper's claims were supported by evidence and the content would be of interest to the TMLR community, and I agree with their assessment.

Please remember to amend the text to conform to the TMLR style (in particular, change red text used for highlighting diffs back to black text) for the publication-ready version.

**Audience:**

All reviewers were in agreement that the paper's contributions would be of interest to the TMLR community. I also agree with their assessment.

**Claims And Evidence:**

All reviewers were in agreement that the paper's claims are supported by adequate evidence. I also agree with their assessment.